# A discrete organoplatinum(II) metallacage as a multimodality theranostic platform for cancer photochemotherapy

Guocan Yu[1], Shan Yu[2], Manik Lal Saha[3], Jiong Zhou [4,5], Timothy R. Cook [5], Bryant C. Yung[1], Jin Chen[2], Zhengwei Mao [2], Fuwu Zhang[1], Zijian Zhou[1], Yijing Liu[1], Li Shao[4], Sheng Wang[1], Changyou Gao[2], Feihe Huang[4], Peter J. Stang[3] & Xiaoyuan Chen [1]

Photodynamic therapy is an effective alternative to traditional treatments due to its minimally invasive nature, negligible systemic toxicity, fewer side effects, and avoidance of drug resistance. However, it is still challenging to design photosensitizers with high singlet oxygen ($^1O_2$) quantum yields (QY) due to severe aggregation of the hydrophobic photosensitizers. Herein, we developed a discrete organoplatinum(II) metallacage using therapeutic cis-$(PEt_3)_2Pt(OTf)_2$ as the building block to improve the $^1O_2$ QY, thus achieving synergistic anticancer efficacy. The metallacage-loaded nanoparticles (MNPs) with tri-modality imaging capability allow precise diagnosis of tumor and real-time monitoring the delivery, biodistribution, and excretion of the MNPs. MNPs exhibited excellent anti-metastatic effect and superior anti-tumor performance against U87MG, drug resistant A2780CIS, and orthotopic tumor models, ablating the tumors without recurrence after a single treatment. Gene chip analyses confirmed the contribution of different therapeutic modalities to the tumor abrogation. This supramolecular platform holds potential in precise cancer theranostics.

---

[1] Laboratory of Molecular Imaging and Nanomedicine, National Institute of Biomedical Imaging and Bioengineering, National Institutes of Health, Bethesda, MD 20892, USA. [2] MOE Key Laboratory of Macromolecular Synthesis and Functionalization, Department of Polymer Science and Engineering, Zhejiang University, Hangzhou 310027, P.R. China. [3] Department of Chemistry, University of Utah, 315 South 1400 East, Room 2020, Salt Lake City, UT 84112, USA. [4] State Key Laboratory of Chemical Engineering, Center for Chemistry of High-Performance and Novel Materials, Department of Chemistry, Zhejiang University, Hangzhou 310027, P.R. China. [5] Department of Chemistry, University at Buffalo, State University of New York, 359 Natural Sciences Complex, Buffalo, NY 14260, USA. Correspondence and requests for materials should be addressed to Z.M. (email: zwmao@zju.edu.cn) or to F.H. (email: fhuang@zju.edu.cn) or to P.J.S. (email: stang@chem.utah.edu) or to X.C. (email: shawn.chen@nih.gov)

Of the many challenging tasks of medicine, none has experienced a more controversial beginning and none has endured a more hard-fought evolution than the treatment of cancer[1–4]. Over the past decades, the understanding of cancer biology has grown tremendously; however, significant improvement to mortality rates for many cancers remains an outstanding goal. With an ever-increasing global cancer burden, the exploitation of effective cancer therapeutic and diagnostic agents is extremely urgent[5–8]. Owing to their severe side effects and tendency to promote drug resistance, chemotherapies remain limited in their clinical applications. In many cases, their efficacies are unsatisfactory and tumor recurrence and metastasis are common[9,10]. In order to enhance patient outcomes, different kinds of treatments may be simultaneously applied to produce additive therapeutic effects by collecting the merits of each treatment possessing distinct anticancer mechanisms[6,11,12].

Photodynamic therapy (PDT), a rapidly developing cancer treatment, combines three non-toxic components (tissue oxygen, light, and photosensitizer) to cause toxicity to tumor cells[13–16]. PDT is an excellent complement to chemotherapy due to its negligible drug resistance, minimal invasion, fewer side effects, and less damage to marginal tissues than occur with conventional cancer treatments. The mechanism of PDT involves energy transfer from the light-excited photosensitizer to molecular oxygen upon irradiation with light at appropriate wavelengths, to generate singlet oxygen ($^1O_2$), resulting in oxidative damage to cancer cells. In sharp contrast to other cancer treatments, such as surgery, chemotherapy, and radiotherapy, PDT can lead to localized destruction of diseased tissues via selective uptake of the PSs and/or local exposure to light, providing a minimally invasive cancer therapy[17,18]. The most extensively used photosensitizers so far are focused on porphyrins and their derivatives, the extended delocalized aromatic macrocycles containing a porphyrin structure——four pyrroles linked by methine bridges. However, the large π-conjugate of planar porphyrin always suffers severe π–π stacking that leads to significant quench of the excited state, thus resulting in the decrease of $^1O_2$ generation quantum yield (QY), greatly limiting their applications in PDT and raising challenges for the development of suitable pharmaceutical formulations[19,20]. Although it is efficient in local ablation of tumors, PDT is incapable of effectively eliminating cancers distal from the primary tumor and infiltrating cancer cells, always leading to tumor recurrence. The combination of systemic chemotherapy and PDT has demonstrated superb efficacy in optimizing the outcome of cancer treatments. Following ablation of the primary tumor, chemotherapeutic drugs may diffuse to a wider range of cancerous cells, thus inhibiting tumor recurrence and eliminating the need for multiple doses.

Herein, we integrate chemotherapy and PDT approaches using an organoplatinum(II) metallacage (M) formed via multicomponent coordination-driven self-assembly using 5,10,15,20-tetra(4-pyridyl)porphyrin (TPP), cis-(PEt₃)₂Pt(OTf)₂ (cPt), and disodium terephthalate (DSTP) as the building blocks (Fig. 1). Upon formation of M, the intermolecular π–π stacking of TPP is effectively suppressed, resulting in significant enhancement of fluorescence and $^1O_2$ generation QY, which is favorable for near-infrared fluorescence imaging (NIRFI) and PDT. Moreover, the porphyrin cores of M are suitable hosts for metal ions, such as Mn and $^{64}Cu$ ions, allowing the implementation of highly effective imaging techniques, such as magnetic resonance imaging (MRI) and positron emission tomography (PET) imaging. PEGylation by mPEG-b-PEBP and RGD-PEG-b-PEBP endows the metallacage-loaded NPs (MNPs) with long blood circulation time and less non-specific cellular uptake via the enhanced permeability and retention (EPR) effect and active targeting ability. Motivated by the efficient tumor uptake of these MNPs revealed by NIRFI, PET, and MRI, in vivo photochemotherapy exhibits superior anticancer outcomes in combating against U87MG, drug-resistant A2780CIS, and orthotopic (breast and hepatoma) tumor models, effectively preventing tumor recurrence and metastasis after a single treatment. Gene chip analyses were also employed to identify the underlying biological contribution of each therapeutic modality and synergistic therapy to achieve tumor abrogation.

## Results

**Fabrication of metallacage-loaded nanoparticles (MNPs).** The molecules and copolymers used here were synthesized and fully characterized by various methods, including gel permeation chromatography (GPC), $^1H$ NMR, and $^{31}P\{^1H\}$ NMR spectroscopies (Supplementary Figures 1–11). For the construction of M, a heteroligation-directed self-assembly methodology was employed[21,22]. The well-defined signals in $^1H$ NMR spectra support the formation of a discrete and highly symmetric assembly as a sole thermodynamic product (Supplementary Figures 12, 13)[23]. Whereas, the $^{31}P\{^1H\}$ spectrum of cPt (Fig. 2a) indicates a single phosphorus environment, and upon assembly, two different doublets at 6.24 and 0.92 ppm with concomitant $^{195}Pt$ satellites were observed in the spectrum of M (Fig. 2b). The observation of two doublets is consistent with the symmetry-breakage that occurs when one carboxylate moiety and one pyridyl coordinates to each Pt center[24]. Electrospray ionization mass spectrometry further supported the formation of M owing to the observation of mass fragments corresponding to intact cores. Two peaks at $m/z = 1158$ and $m/z = 1486$ were monitored (Supplementary Figure 14), ascribed to the intact tetragonal prism core with charge states, $[M − 5OTf]^{5+}$ and $[M − 4OTf]^{4+}$, respectively. The isotopic resolution of each peak was in agreement with the corresponding theoretical isotopic distribution, indicating that M possessed the expected 1:2:4 ratio of building blocks.

With the MNPs in hand, their physicochemical properties were investigated to verify that the $^1O_2$ generation QY was dramatically enhanced by the polynuclear metallacage. As shown in the UV-vis spectra of TPP and M (Supplementary Figure 15) collected in 1:1 dichloromethane/methanol, TPP shows a major Soret peak at 445 nm and four Q-bands at 519, 555, 586, and 640 nm. These bands show a minor redshift upon assembly into a cofacial arrangement in M, a potential ramification of interactions between the π systems of the two macrocyclic PSs. The Soret absorption of M appears at 450 nm, while the Q-bands are observed at 522, 557, 596, and 652 nm. However, the coordination-triggered self-assembly was insufficient to completely suppress aggregation of M in water, because the solubility of M was still poor. Indeed, only slight fluorescence enhancement was observed by the formation of M as compared with free TPP.

In order to effectively improve the solubility and stability of M under physiological condition while increasing its biocompatibility and bioavailability, mPEG-b-PEBP and RGD-PEG-b-PEBP were utilized to encapsulate M into NPs with a loading content of 46.4% (Supplementary Table 1). This nano-formulation endowed the resultant M-loaded NPs with excellent $^1O_2$ generation QYs, long blood circulation time, low side effects, and high accumulation in tumors through EPR effect and active targeting ability[25–28]. The size and morphology of MNPs were investigated by both atomic force microscopy (AFM) and dynamic laser scattering (DLS). As shown in Fig. 2c, spherical NPs with diameters ranging from 40 to 80 nm were observed, in line with the DLS measurement (Fig. 2d). An increase in diameter from 31.4 to 63.6 nm was found after loading M into the NPs, suggesting the successful encapsulation of M by mPEG-b-PEBP

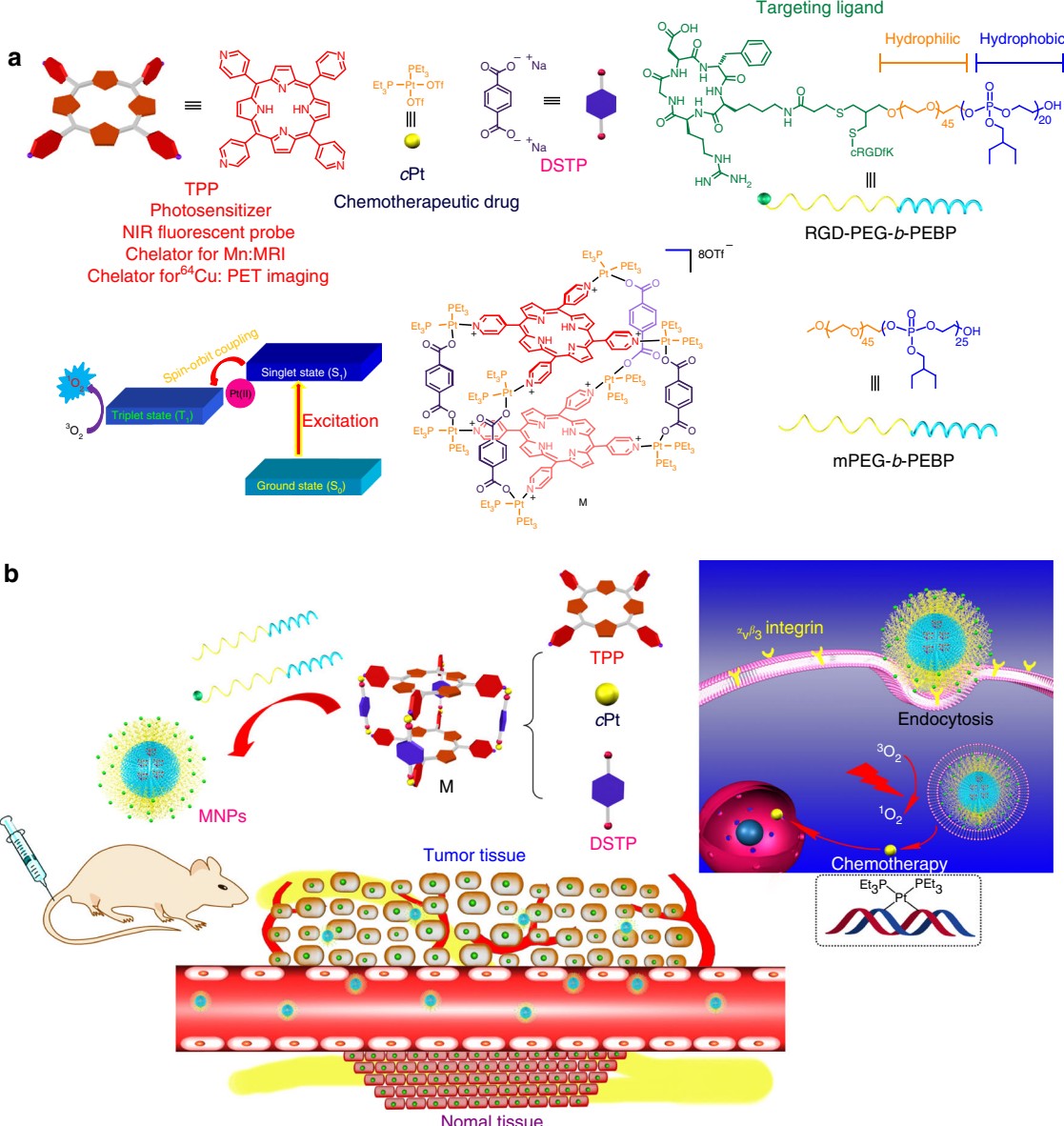

**Fig. 1** Schematic diagrams of the MNPs serving as a multifunctional theranostic platform. **a** Structures of TPP, *c*Pt, DSTP, M, mPEG-*b*-PEBP, and RGD-PEG-*b*-PEBP. **b** Schematic illustration of MNPs accumulation in tumor tissue followed by EPR effect and receptor-mediated endocytosis, and their applications in subcutaneous (U87MG), drug-resistant (A2780CIS), orthotopic (4T1 and LM3) tumor treatments, and lung anti-metastasis

and RGD-PEG-*b*-PEBP. Additionally, the zeta potential of the NPs formed from mPEG-*b*-PEBP and RGD-PEG-*b*-PEBP increased from −57.9 mV to −2.2 mV upon encapsulation of M (Fig. 2e). Solutions containing MNPs in phosphate-buffered saline (PBS) containing 10% fetal bovine serum at 37 °C were stable over 72 h of incubation and negligible changes in size was found (Supplementary Figure 16), demonstrating the high colloidal stability of MNPs in biological buffer. In the MNPs, the hydrophobic PEBP chains intertwined with M to form the cores, whereas the PEG chains occupied the external faces to interact with water as the shells of the stable MNPs.

Singlet oxygen sensor green (SOSG) was utilized to detect the $^1O_2$ produced by TPP, M, and MNPs upon laser irradiation. As shown in Fig. 2f, negligible changes in fluorescence intensity at 532 nm were observed for TPP and M caused by the attenuation of $^1O_2$ generation ascribing to their poor solubility and π–π stacking interactions. In sharp contrast, the SOSG fluorescence exhibited a significant enhancement upon introduction of MNPs

under the same condition, suggesting a high photosensitizing efficacy (Fig. 2f and Supplementary Figure 17). When the well-known $^1O_2$ scavenger sodium azide (NaN$_3$) was added, the fluorescence changes at 532 nm were inhibited, further confirming the production of $^1O_2$ during laser irradiation (Fig. 2g). By using 1,3-diphenyl isobenzofuran (DPBF) as a $^1O_2$ indicator and *meso*-tetrakis(p-sulfonatophenyl) porphyrin tetrasodium salt (TPPS) as a standard ($^1O_2$ generation QY = 0.60)[29], the $^1O_2$ generation QYs of the MNPs were calculated to be 0.44 in water (Supplementary Figure 18), around 110-fold higher than that of TPP, demonstrating that the formation of metallacage played a significant role in improving the $^1O_2$ generation QY. Time-dependent density functional theory molecular simulation was conducted to provide theoretical support for its extremely high $^1O_2$ generation QY value (Fig. 2h and Supplementary Figure 19). The energy gap between the lowest singlet excited state and the lowest triplet excited state of M was quite small, paving the way for increasing the intersystem crossing rate[30], and consequently

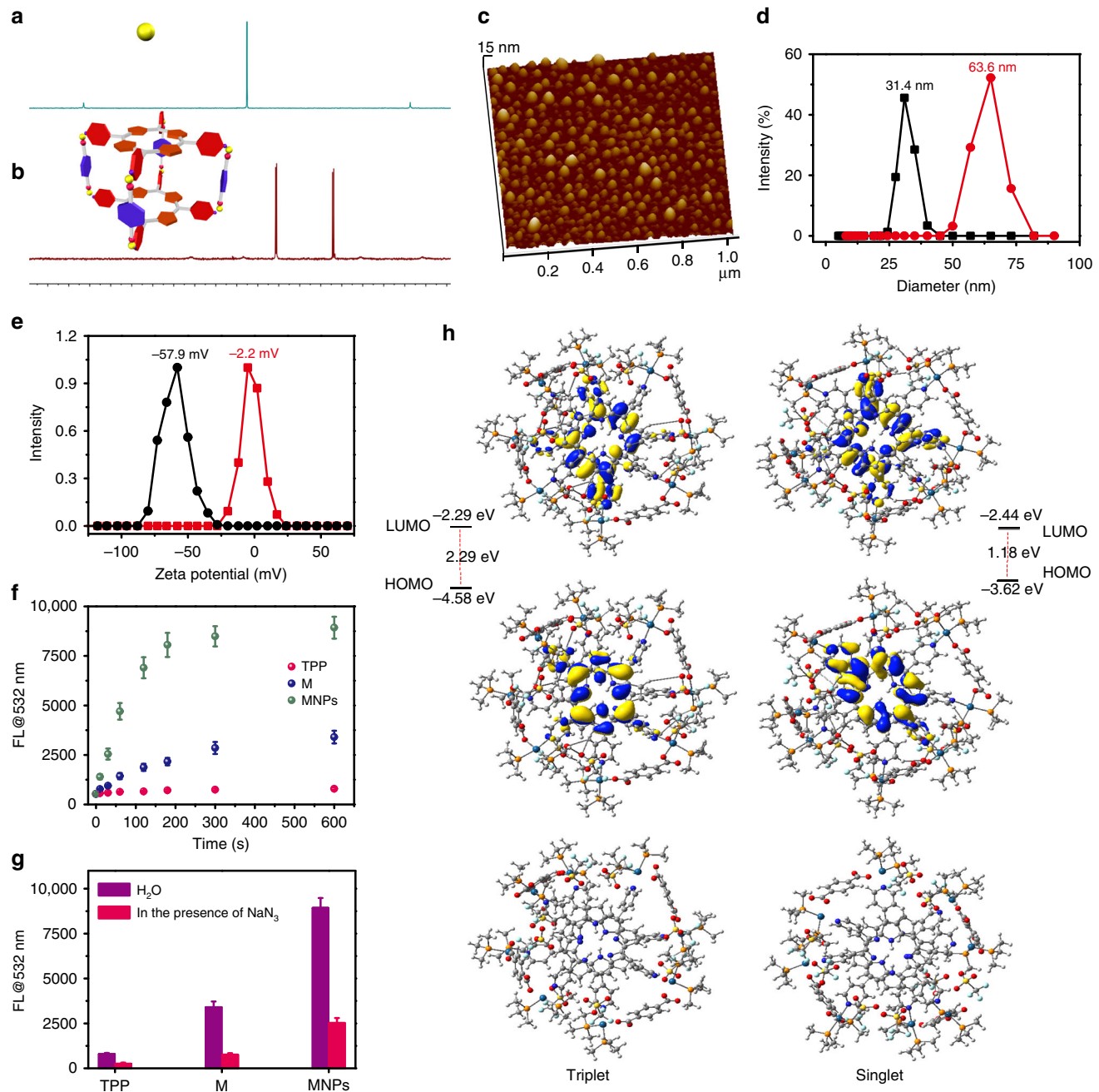

**Fig. 2** Characterizations of the MNPs and their PDT effect. $^{31}P\{^{1}H\}$ (202.3 MHz, 293 K) of **a** cPt and **b** M. **c** AFM image of the MNPs. **d** DLS and **e** Zeta potential results of the NPs formed from mPEG-*b*-PEBP and RGD-PEG-*b*-PEBP before (black line) and after (red line) encapsulation of M. **f** The fluorescence intensity changes at 532 nm of the solution containing SOSG and TPP (M or MNPs) after different periods of irradiation (671 nm, 0.5 W cm$^{-2}$). **g** The fluorescence intensity changes at 532 nm of the solution containing SOSG and TPP (M or MNPs) after irradiation (671 nm, 0.5 W cm$^{-2}$, 10 min) in the absence and presence of NaN$_3$ (10.0 μM). **h** Time-dependent density functional theory molecular simulation of M. The data are expressed as means ± s.d

improving the generation of $^{1}O_2$. The incorporation of heavy atoms (Pt) into M further promoted $^{1}O_2$ generation, because Pt possessed a high spin–orbit coupling constant ($\chi = 4481$ cm$^{-1}$) enhancing the rapid intersystem crossing of singlet-to-triplet with a high rate ($10^{12}$ s$^{-1}$)[31,32].

**In vitro synergistic anticancer treatments.** The $\alpha_v\beta_3$ integrin receptor, one of the most-specific markers of tumor vasculature, is low expressed on mature endothelial cells or epithelial cells, but is often overexpressed on many tumor cells, such as

osteosarcoma, melanoma, glioblastoma, and carcinomas of the lung and breast[33]. Cyclo(Arg-Gly-Asp-D-Phe-Lys) (cRGDfK) has been selected as a specific targeting ligand because it selectively binds to $\alpha_v\beta_3$ integrin with high affinity, endowing the MNPs with excellent targeting ability[34,35]. Receptor-mediated endocytosis of the MNPs was confirmed by confocal laser scanning microscopy (CLSM), flow cytometry, inductively coupled plasma mass spectrometry (ICP-MS), and 3-(4′,5′-dimethylthiazol-2′-yl)-2,5-diphenyl tetrazolium bromide (MTT) assay on an $\alpha_v\beta_3$ integrin overexpressing U87MG cell line. Red fluorescence arising from the MNPs was detected in the cell cytoplasm after 2 h

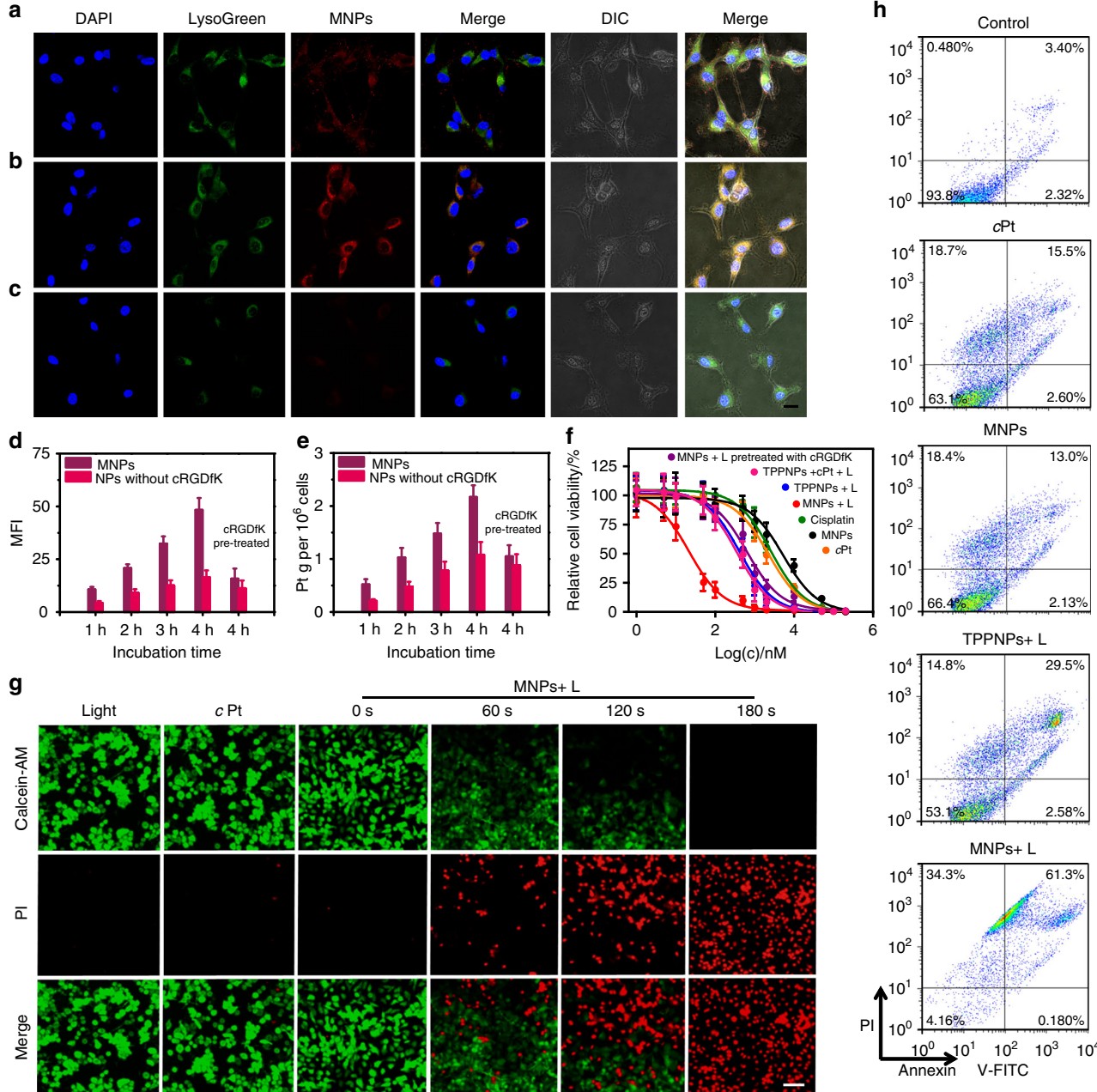

**Fig. 3** In vitro targeted delivery activity of the MNPs and synergistic anticancer effect. CLSM images of U87MG cells cultured with MNPs for **a** 2 h, **b** 4 h, and **c** 4 h pre-treated with free cRGDfK for 30 min. Scale bar is 20 μM. **d** Mean fluorescent intensity (MFI) and **e** intracellular platinum amount of U87MG cells after treatment with the MNPs or NPs without cRGDfK for different incubation times. **f** Cytotoxicity of U87MG cells treated with different administrations. **g** Fluorescence images of U87MG cells co-stained with Calcein AM/PI after different treatments. The platinum concentration of cPt and MNPs was kept at 200 nm. Scale bar is 50 μM. **h** FCM analysis of U87MG cell apoptosis induced by different treatments. The irradiation density was 0.1 W cm⁻² at 671 nm, and the irradiation time was 3 min. The data are expressed as means ± s.d

incubation (Fig. 3a) and the fluorescence signal was much brighter when the incubation time was extended to 4 h (Fig. 3b), indicating effective internalization of the MNPs. The red fluorescence signal overlapped well with the green fluorescence from LysoTracker green, indicating that the MNPs were uptaken by the cells and eventually trapped inside lysosomes. Pre-treatment of free cRGDfK (20 μM) for 30 min greatly weakened the intracellular red fluorescence due to the blockage of the integrin on the cell membrane (Fig. 3c). Compared with the NPs without cRGDfK self-assembled from M and mPEG-*b*-PEBP, FCM and

ICP-MS studies showed that U87MG cells exhibited a higher uptake rate and intracellular accumulations of the MNPs at various incubation time points (Fig. 3d, e). Additionally, pre-treatment with 20 μM of free cRGDfK for 30 min resulted in inefficient cellular uptake of MNPs, the intracellular platinum content decreased from 2.17 to 1.05 ng/10⁶ cells after 4 h incubation with MNPs (Fig. 3e). These experiments firmly demonstrated the $\alpha_v\beta_3$ integrin receptor-mediated uptake of the MNPs, implying the presence of targeting ligands facilitated their cellular internalization.

The intracellular generation of $^1O_2$ by MNPs after cancer cell uptake upon light irradiation (671 nm, 0.1 W cm$^{-2}$) was measured by using 2′,7′-dichlorofluorescein diacetate (DCFDA) as the indicator. Negligible signal from DCF was observed when the cells were cultured with DCFDA or the MNPs without light irradiation; however, strong green fluorescence of DCF was visible inside the cells upon light irradiation for 3 min in the presence of MNPs because of the efficient generation of $^1O_2$ (Supplementary Figure 20). DCF fluorescence was dramatically inhibited for the cells treated with a $^1O_2$ scavenger vitamin C, further providing convincing evidence for the generation of $^1O_2$. A control experiment used TPPNPs fabricated from TPPN, mPEG-*b*-PEBP and RGD-PEG-*b*-PEBP for photodynamic therapy. The green signal arising from DCF was much weaker for the cells cultured with TPPNPs than for those treated with MNPs at the same concentration of TPP (Supplementary Figure 20). Indeed, the $^1O_2$ generation QY value of TPPNPs was determined to be 0.12 by using TPPS as a standard (Supplementary Figure 18), emphasizing that the formation of metallacage significantly enhanced its $^1O_2$ generation QY.

Through a combination of chemotherapy and PDT into a single platform, the MNPs efficiently mediated both apoptosis and necrosis when irradiated, thus achieving a synergistic anticancer effect. MTT assay indicated that the half-maximal inhibitory concentration (IC$_{50}$) of the photochemotherapy (MNPs + L) was much lower than those of single chemotherapy (cisplatin, *c*Pt and MNPs), single PDT (TPPNPs + L) and even the mixture of *c*Pt and TPPNPs (4:1 in molar ratio) upon laser irradiation (Fig. 3f), underlining that the formation of M played a remarkable role in the synergistic photochemotherapy. The enhanced cytotoxicity of the MNPs resulted from the excellent synergistic effect of chemotherapy and PDT as judged by the combination index (CI) value at IC$_{50}$ that was much lower than 1 (CI = 0.11), where *c*Pt formed intra- and inter-strand crosslinks on DNA through coordination of the N$_7$ atoms on the purine bases (Supplementary Figures 21–23), while PDT caused cytotoxicity through both apoptosis and necrosis pathways by oxidizing the intracellular DNA or proteins[36–38]. The phototoxicity index defining the ratio of the IC$_{50}$ values in dark and exposure to light irradiation was calculated to be as high as 246 (Fig. 3f). The extremely high phototoxicity index of the MNPs suggested that they are excellent phototherapeutic agents. The cytotoxicity of the photochemotherapy was greatly diminished by pre-treating with free cRGDfK for 30 min, arising from the decreased internalization of the MNPs. Calcein AM/propidium iodide (PI) co-staining indicated that negligible cell death was induced by light, *c*Pt, and MNPs due to the extremely low concentrations (Fig. 3g). Compared with these control groups, significantly increased cell death was observable after incubating with MNPs followed by light exposure. The percentage of the dead cells increased effectively accompanied with the extension of laser exposure time, and it took 180 s to kill all the cells, confirming that MNPs possessed excellent photodynamic therapeutic effect.

Annexin V-FITC/PI assay was employed to distinguish viable cells from dead cells of different phases using flow cytometry, which were identified as viable, early apoptotic, late apoptotic, and necrotic cells, respectively (Fig. 3h). For the cells treated with *c*Pt, the populations of the early apoptotic, late apoptotic, and necrotic cells were 2.60%, 15.5%, and 18.7%, respectively, close to those of MNPs without irradiation. For the PDT (TPPNPs + irradiation) at the light dose of 0.1 W cm$^{-2}$ for 3 min, the percentage of the cells at late apoptotic and necrotic stage increased to 29.5% and 14.8%, respectively. Excitingly, the cell mortality rate significantly increased to 95.6% (61.3% for later apoptosis and 34.3% for necrosis) combining chemotherapy and

PDT, confirming the promising application of the MNPs in synergistic treatment against cancer.

**In vivo tri-modality imaging**. The simultaneous use of highly sensitive and high-resolution multimodal imaging methods helps to overcome the limitations of each modality independently and offers complementary and accurate insights into tumor characteristics, which are highly desired in cancer diagnostics[6,39]. Among multimodal imaging, the combination of PET imaging, MRI, and NIRFI is the most explored trimodal scheme[40,41]. PET and MRI are complementary to each other given the high sensitivity of PET and the unparalleled spatial resolution of MRI. Additionally, NIRFI further validates, correlates, or adds new information to the observations at a low cost. However, it remains a huge challenge to integrate different imaging contrasts into one platform to achieve multi-modality imaging because of the complicated synthetic procedures and the potential interference between the contrasts.

As mentioned above, the fluorescence intensity of the porphyrin prisms increased in the MNPs, indicating that deleterious π–π stacking was suppressed (Fig. 4a). The intense emission provided the basis for use as an optical probe for NIRFI. Figure 4e shows that significant tumor accumulation was clearly visible in the mice administered with MNPs at 6 h post injection, and exceptionally intensive signal was visible in the tumor area for more than 24 h in comparison with other tissues. This high tumor specificity was attributed to the outstanding active targeting and EPR effects of the MNPs. To collect more accurate information about the fluorescence intensities within tissues, the mice were killed, ex vivo NIRFI of isolated organs (heart, liver, spleen, lung, kidneys, intestine, and tumor) was conducted at 24 h post injection (Fig. 4f). The excised tumor showed notable fluorescence intensity, whereas the lung, spleen, and heart had very low signals.

Porphyrins are well-known for their high affinity toward metal ions, such as Fe, Co, Cu, Ni, Zn, and Mn. Depending on the metal used, a variety of functions are enabled, including use as imaging agents[42,43]. The successful introduction of $^{64}Cu$ into the MNPs was verified using energy dispersive spectrometry (EDS) by using non-radioactive Cu, where the characteristic peaks related to Cu were detected in the spectrum (Supplementary Figure 24). Determined by instant thin layer chromatography (iTLC) and size-exclusion centrifugation filtration, the labeling achieved $^{64}Cu@MNPs$ with a radiochemical yield of ~100% (Supplementary Figure 25). The radiolabeling procedure attached a $^{64}Cu$ label to <5% of the porphyrin in the MNPs without altering their size, photonic properties and in vivo circulation behaviors. Ascribing to the high affinity between $^{64}Cu$ and TPP, the stability of chelation in mouse serum was determined to be 98.6 ± 0.9% after 24 h incubation (Fig. 4b), highlighting that the in vivo PET imaging accurately revealed the behavior of the $^{64}Cu@MNPs$ rather than the cleaved $^{64}Cu$.

The incorporation of PET properties into normal optical probes addresses a series of limitations faced by fluorescence imaging. For example, PET offers non-invasive imaging with deep tissue penetration and quantitative biodistribution of the MNPs that can hardly be achieved by fluorescence imaging. Encouraged by the quantitative radiolabeling yield of MNPs by isotope $^{64}Cu$, in vivo PET imaging was conducted to track the delivery and biodistribution of $^{64}Cu@MNPs$ using a U87MG xenograft model (Fig. 4g). Quantitative region of interest analysis on the whole-body images showed the tumor uptake of $^{64}Cu@MNPs$ was 0.81 ± 0.07, 3.23 ± 0.38, 5.63 ± 0.64, 7.36 ± 0.86, 9.41 ± 1.07, and 6.68 ± 0.82% ID g$^{-1}$ at 2, 4, 6, 12, 24, and 48 h after i.v. injection, respectively (Fig. 4i). On the other hand, the dynamic

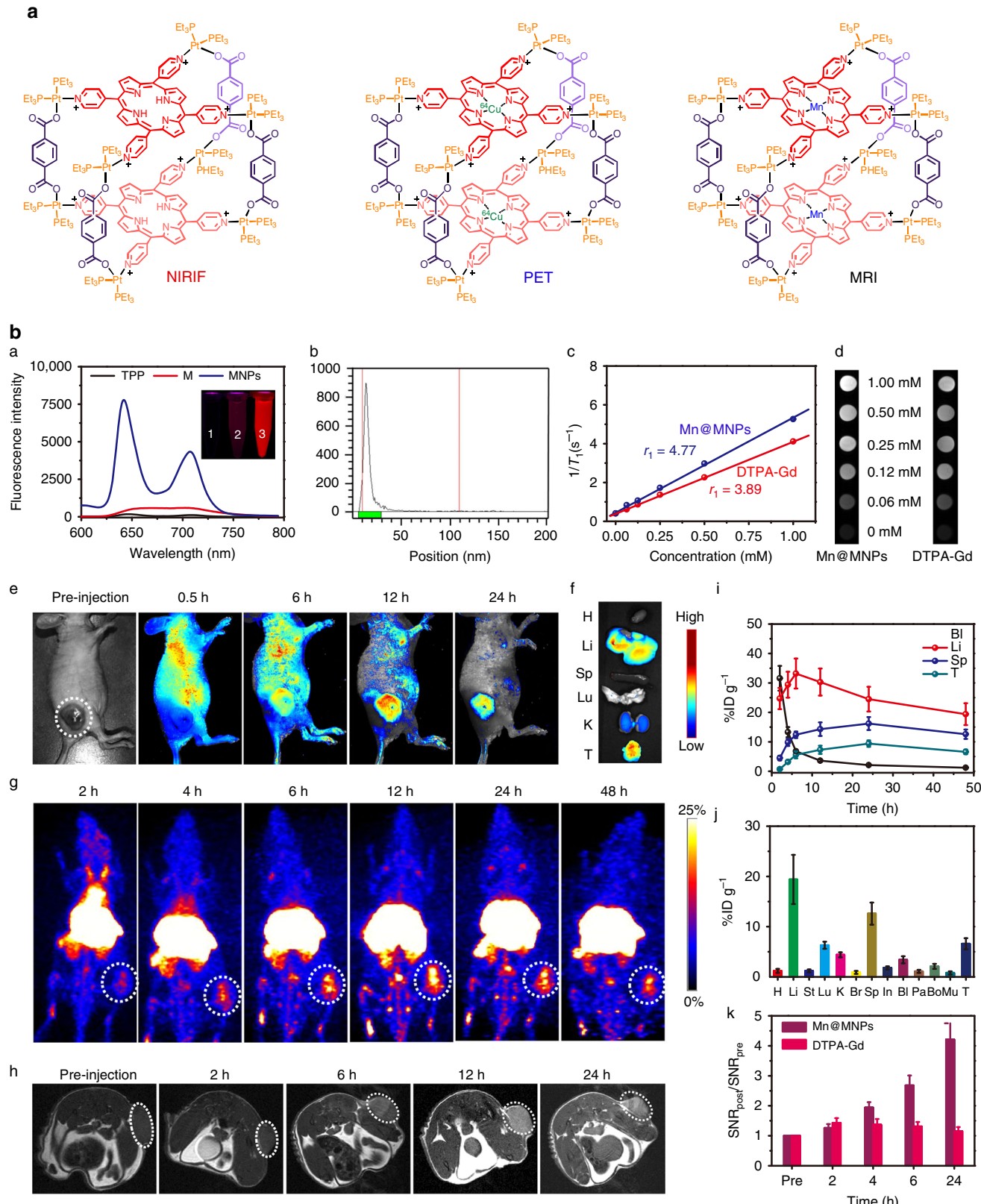

accumulations of $^{64}$Cu@MNPs in liver, spleen, and bladder were also detected during the observation (Fig. 4i). To verify the accuracy of PET quantitative investigations, ex vivo biodistribution analysis of $^{64}$Cu radioactivity in major organs was conducted through $\gamma$ counting (Fig. 4j). The average heart-, liver-, spleen-, lung-, kidney-, and tumor-to-muscle ratios were calculated to be

$1.54 \pm 0.38$, $23.6 \pm 4.7$, $15.8 \pm 4.4$, $7.88 \pm 1.16$, $5.52 \pm 0.77$, and $8.25 \pm 1.03$, respectively, at 48 h post injection. Compared with the data obtained from PET and ICP-MS investigations, the results concerning the biodistribution of the MNPs was consistent, demonstrating that non-invasive PET imaging reflected the distribution of the MNPs accurately.

**Fig. 4** In vivo tri-modality imaging. **a** Fluorescence spectra of TPP, M, and MNPs. Inset picture is the fluorescent imaging related to the solution (1, TPP; 2, M, 3, MNPs). **b** Radio TLC chromatograms of $^{64}Cu@MNPs$ after 24 h incubation in mouse serum. **c** Plots of $T_1^{-1}$ versus Mn (or Gd) concentration for Mn@MNPs (or DTPA-Gd). **d** $T_1$-weighted MRI results (7T) obtained from aqueous solution of Mn@MNPs (or DTPA-Gd) at various Mn (or Gd) concentrations. **e** NIRFI of U87MG tumor-bearing nude mice following i.v. injection of MNPs. The white circle denotes the tumor site. **f** Ex vivo image of the main organs separated from U87MG tumor-bearing mice at 24 h post injection of MNPs. **g** PET image of U87MG tumor-bearing nude mice at 2, 4, 6, 12, 24, and 48 h post injection of $^{64}Cu@MNPs$ (150 μCi). The white circle denotes the tumor site. **h** In vivo $T_1$-weighted axial MRI images (7T) of the mice pre-injection and after injection of Mn@MNPs. The white circle denotes the tumor site. **i** Time-activity curves of the biodistribution of $^{64}Cu@MNPs$ in the blood, liver, spleen, and tumor ($n = 3$). **j** Quantitative biodistribution of $^{64}Cu@MNPs$ in the main organs 48 h post injection of $^{64}Cu@MNPs$. H heart, Li liver, St stomach, Lu lung, K kidney, Br brain, Sp spleen, In intestine, Bl bladder, Pa pancreas, Bo bone, Mu muscle, T tumor. **k** Quantificational analysis of SNR ratio in tumor sites at 2, 4, 6, and 24 h post injection of Mn@MNPs or DTPA-Gd ($n = 3$). The data are expressed as means ± s.d

The potential renal toxicity of gadolinium (Gd)-based MRI agents greatly limits their clinical applications[44]. Considering their lower intrinsic toxicity, relatively high electronic spin, and fast water exchange rates, manganese (Mn)-based MRI agents have been regarded as prominent alternatives[45,46]. Apart from Cu, the MNPs also have the intrinsic ability to chelate paramagnetic Mn to generate a contrast agent for MRI. Similarly, the result obtained from EDS supported that Mn was incorporated into the MNPs (Supplementary Figure 26). The $T_1$-weighted MR phantom of Mn@MNPs was examined at 7.0 T magnetic field. The longitudinal relaxivity ($r_1$) of Mn@MNPs was calculated to be $4.77 \pm 0.56\ mM^{-1}\ S^{-1}$ by plotting the inverse relaxation time against the Mn concentration (Fig. 4c, d), which appeared to be higher than that of the clinically approved $T_1$-contrast agent Magnevist (DTPA-Gd) ($3.89 \pm 0.40\ mM^{-1}\ S^{-1}$). Different from the traditional hydrophobic polymers, the high flexibility and hydration of the polyphosphoester backbones were favorable for water to access to the inner Mn ions that are otherwise obscured within the hydrophobic core, thus maintaining the high $r_1$ value of the MRI agent by shortening their spin-lattice relaxation time.

In vivo $T_1$-weighted MRI of tumor-bearing mice was carried out by i.v. administration of Mn@MNPs. The dramatic whitening effect in the tumor area was observed and the MR signal enhanced over time. The tumor area exhibited remarkable $T_1$ contrast from the surrounding normal tissues after 6 h post injection (Fig. 4h). Quantitative analysis showed that the signal-to-noise ratio (SNR) increased up to $1.26 \pm 0.13$ at 1 h post injection and maintained as high as $4.21 \pm 0.54$ at 24 h post injection owing to the EPR effect and active targeting ability of Mn@MNPs, whereas the SNR value of Gd-DTPA was around 1.41 at 2 h post injection, and then diminished rapidly caused by its fast clearance (Fig. 4k).

**In vivo synergistic photochemotherapy.** The pharmacokinetics of MNPs and cPt were studied by quantifying the blood Pt concentration using ICP-MS. The blood circulation half-life of the MNPs was calculated to be $2.18 \pm 0.33$ h (Fig. 5a and Supplementary Figure S27), 8.07 times as that of cPt ($0.27 \pm 0.04$ h). Approximately 3.79 and 1.48% of the injected dose (ID) remained in the plasma at 24 and 48 h post injection for the MNPs, respectively. While cPt was considerably eliminated from the bloodstream, showing only 1.61 and 0.52% of the ID existed in the plasma at 2 and 4 h post injection (Fig. 5a and Supplementary Figure 28). The area under the curve of the MNPs was 19.8-fold greater than that of cPt, suggesting that the circulation time was greatly elongated through nano-formulation attributing to the EPR effect and active targeting.[47] To acquire a quantitative assessment of biodistribution of the MNPs and free cPt, organs were digested and analyzed for Pt content by ICP-MS (Fig. 5b, c). Biodistribution evaluations indicated that the MNPs effectively accumulated in tumor, greatly avoiding capture by the RES, as evidenced by the relatively low Pt amount in the liver and spleen.

For example, tumor accumulation of the MNPs approached $1.82 \pm 0.17\ μg\ g^{-1}$ tissue at 24 h post administration, much higher than that of cPt ($0.47 \pm 0.06\ μg\ g^{-1}$ tissue).

Encouraged by the outstanding in vitro synergistic efficacy, excellent pharmacokinetics and biodistribution results of the MNPs, in vivo anti-tumor efficacy and systemic toxicity of the MNPs were evaluated after a single treatment. The nude mice bearing U87MG tumor were randomly divided into seven groups, and formulated with PBS, light (L), cisplatin, cPt, TPPNPs + L, MNPs, or MNPs + L ($n = 9$/group). Compared with the mice treated with PBS, a similar tumor growth was monitored in the light-treated group (Fig. 5d), suggesting that laser irradiation had a negligible influence on the tumor growth. For the mice treated with cisplatin or cPt, the growth of tumors was slightly reduced. The poor anti-tumor efficacy of these two small-molecular-weighted drugs was ascribed to their fast clearance in bloodstream, leading to low accumulation of chemotherapeutic drugs in tumor sites. MNPs-mediated chemotherapy inhibited around half of the tumor growth, more efficient than free cisplatin or cPt, which could be explained by the increased tumor accumulation of the MNPs arising from their extended circulation time, EPR effect, and active targeting capability. It should be noted that although the tumor growth was notably inhibited by the PDT in the first 9 days using TPPNPs followed by irradiation, a rapid recovery of tumor growth was evident afterwards (Supplementary Figure S29), because single PDT hardly cleared all cancer cells distal from the primary tumor and infiltrating cancer cells. Excitingly, the mice treated with MNPs + L displayed the highest anti-tumor efficiency among these groups and completely eradicated tumors within the entire experimental period without recurrence. The tumor growth inhibition ratio of the groups treated with MNPs upon light irradiation was 100%, while that for light, cisplatin, cPt, MNPs, and TPPNPs + L was 6.15%, 32.0%, 48.3%, 70.3%, and 86.2%, respectively (Fig. 5e and Supplementary Figure 30). These results definitely demonstrated that combined anti-tumor efficacy between MNPs-mediated chemotherapy and laser irradiation-active PDT ablated the tumor completely without any recurrence after a single treatment.

Haematoxylin and eosin (H&E) staining revealed that no signs of tissue damage were detected for the mice treated with PBS or light (Fig. 5g). Stained tumor sections from the mice received chemotherapy (cPt, cisplatin, and MNPs) or PDT (TPPNPs + L) showed varying level of apoptotic damage in comparison with PBS-treated group, indicating that these administrations had anti-tumor effect to different extent. In the mice treated with MNPs + L, the H&E staining indicated severe necrosis and significant tissue loss across a large tumor area at 2 days post injection, evidencing its successful destruction of tumor. Transferase-mediated dUTP nick end-labeling (TUNEL) and Ki67-positive immunohistochemical staining further revealed the highest apoptotic levels (green spots) and the lowest proliferation (brown spots) in the tumor from the photochemotherapy group (Fig. 5g and Supplementary Figure 31).

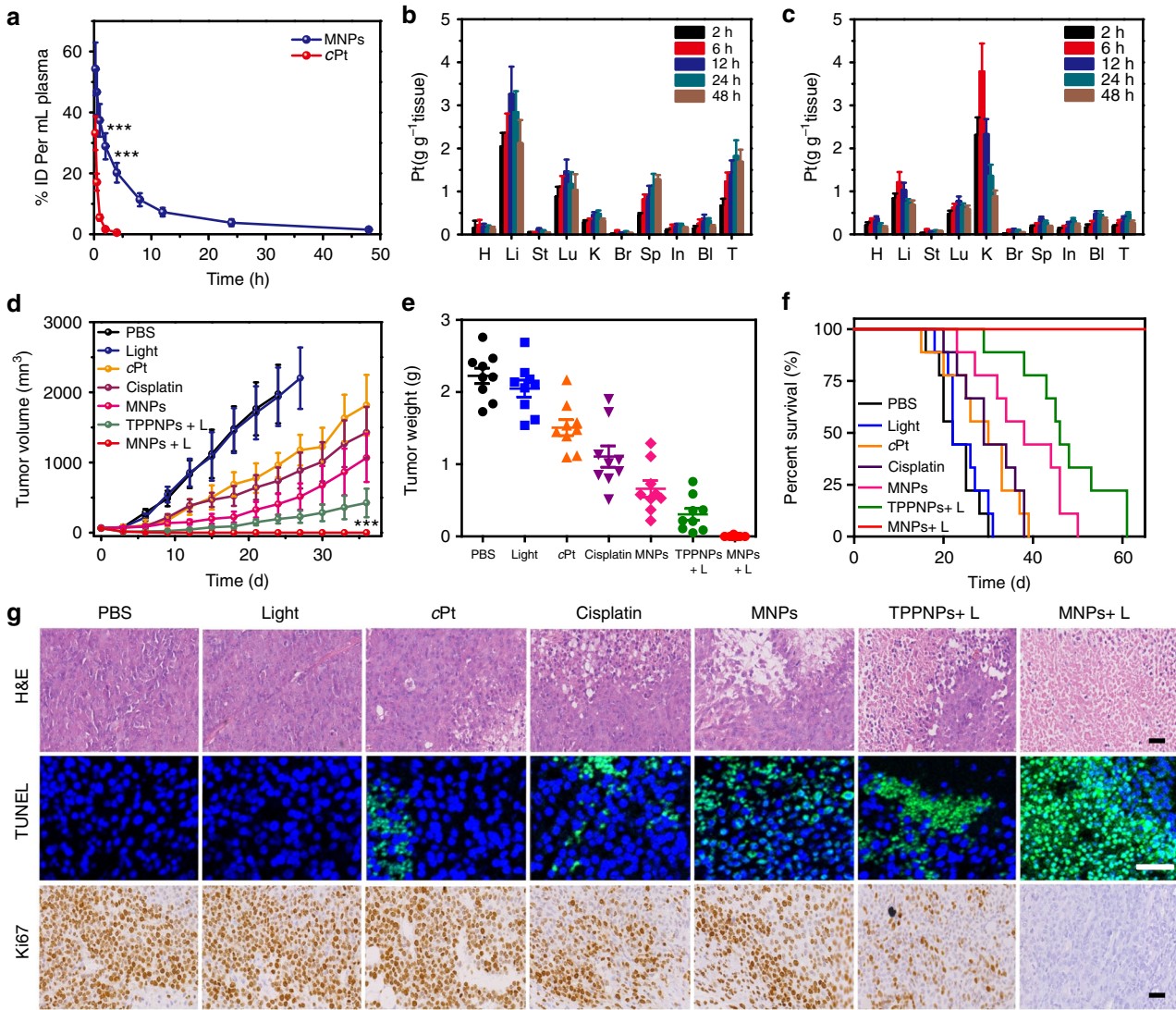

**Fig. 5** In vivo synergistic anti-tumor effect. **a** Plasma platinum concentration versus time after injection of *c*Pt and the MNPs (2 mg Pt per kg) (*n* = 4). Biodistributions of Pt at different time points after injection of **b** MNPs and **c** *c*Pt (2 mg Pt per kg) (*n* = 4). H heart, Li liver, St stomach, Lu lung, K kidney, Br brain, Sp spleen, In intestine, Bl bladder, T tumor. **d** Tumor growth curves for the mice after different formulations (*n* = 9). **e** Weight of U87MG tumors from the mice received different formulations. **f** Kaplan–Meier plots of the mice bearing U87MG tumors after treatment with different formulations. The irradiation density was 0.3 W cm$^{-2}$ at 671 nm, and the irradiation time was 10 min. **g** H&E, TUNEL, and Ki67 staining of tumor tissues collected from the mice administrated with various formulations. Scale bar is 100 μM. The data are expressed as means ± s.d., ***$P < 0.001$

The systemic toxicity of the nanomedicine was a crucial issue for practical applications, which was carefully evaluated using body weight loss and survival rate as indications. For *c*Pt and free cisplatin administrations, the body weight of mice decreased during the first week due to their severe side effects (Supplementary Figure 32). Nephrotoxicity, pulmonary, and hepatic damage were observable at different levels in normal tissues from the mice treated with *c*Pt or cisplatin (Supplementary Figure 33). However, no apparent changes in body weight were observed for the mice received MNPs-based chemotherapy and photochemotherapy (MNPs + L) during the treatment, implying that systemic toxicity was reduced in these mice. Importantly, H&E staining showed no apparent morphological changes or tissue damage in the normal tissues between the healthy mice and the mice after administration of the MNPs. The median survival rate for the mice treated with PBS, light, *c*Pt, cisplatin, MNPs, and TPPNPs + L were calculated to be 22, 22, 30, 29, 38, and 46 d, respectively, whereas the photochemotherapy greatly prolonged mice survival over 60 days without a single death (Fig. 5f). The long-term potential

toxicity of the MNPs in vivo was further evaluated by blood biochemical examination, which indicated that the markers all fell within normal ranges (Supplementary Figures 34, 35), showing no noticeable toxicity and inflammatory response. These results demonstrated that the combination of chemotherapy and PDT in the tumor treatment effectively improved the survival quality of mice and prolonged their lifetime.

**Gene expression analysis of tumors**. After evaluation of in vivo therapeutic performance, we investigated the whole-genetic changes of treated U87MG xenograft tumors to reveal its response to various treatments, including chemotherapy, PDT, and photochemotherapy[48–50]. Compared with untreated group, the chemotherapy group had 302 differential genes (27 upregulated and 275 downregulated), the PDT group had 522 differential genes (130 upregulated and 392 downregulated) and the photochemotherapy group had 2313 differential genes (928 upregulated and 1385 downregulated) (Fig. 6a). The volcano plots

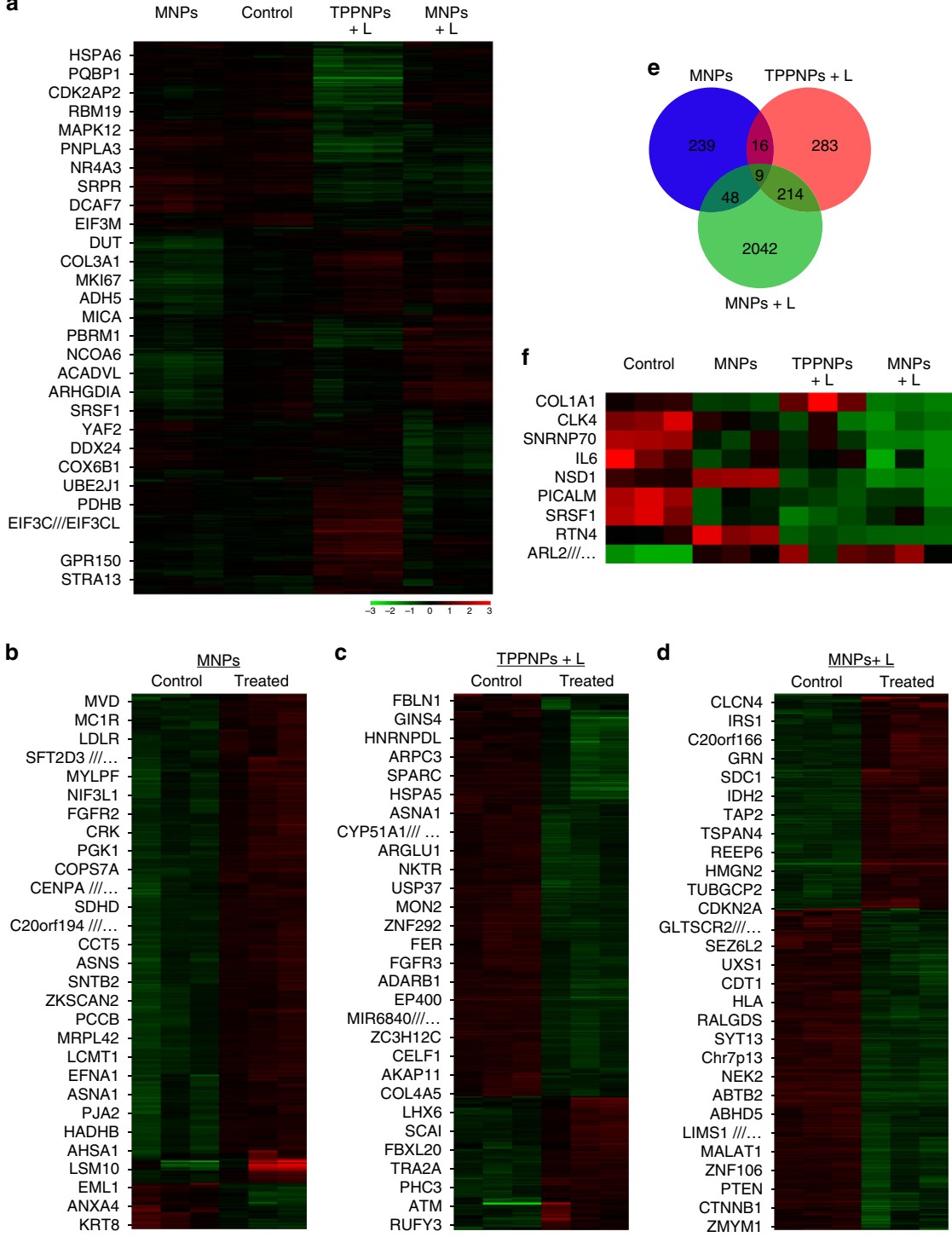

**Fig. 6** Whole-gene expression analyses of tumors from the mice treated with different formulations. **a** Heat-map for GeneChip® PrimeView™ Human Gene Expression Array of mice treated with chemotherapy, PDT or photochemotherapy. Altered genetic profile of tumors treated with **b** MNPs, **c** TPPNPs + L, or **d** MNPs + L compared with the untreated control group. **e** The venn diagram that displays the number of common characteristics among significantly altered genes from three different treatments. Arabic numerals represent the number of significantly altered genes. **f** Selection of potential gene targets involved in tumor regression in response to different therapies

for the different therapeutic groups were obtained by plotting the negative log of the *p*-value on the *y*-axis and the log2 fold change value on the *x*-axis. Highly deregulated genes (black dots) appeared further to the left and right sides, while highly significant changes (color dots) appeared higher on the plot (Supplementary Figures 36–38). A Pearson correlation coefficient

matrix was generated to identify the relationship between the three therapeutic groups and untreated control group (Supplementary Figures 39–42)[51]. This matrix indicated that all the replicates correlated very well (the diagonal of the heat map). Photochemotherapy group showed the most different gene expression profile compared to the control group, while

chemotherapy and PDT groups showed relatively smaller difference comparing with the control group.

A hierarchical clustering of these differentially expressed gene sets, with a fold-change cutoff of ≥1.5, was generated to determine which genes are up- or downregulated after different therapies, according to a previously reported method[52]. Based on this analysis, we summarized the top 25 largest absolute fold-change genes from the three therapeutic groups for further analysis (Supplementary Figures 43–45). For example, the depletion of a cluster of genes was found in response to chemotherapy (Supplementary Figure 43), which are associated with cytoskeleton organization, such as SDCBP (syndecan binding protein), ACTA1 (actin, alpha 1), ACTG2 (actin, gamma 2), TPM1 (tropomyosin 1 alpha), and MYLPF (myosin regulatory light chain 2). Besides, several genes related to cell proliferation or cell cycle, including MAD2L1 (AD2 mitotic arrest deficient-like 1) and HNRNPF (heterogeneous nuclear ribonucleoprotein F), were also downregulated in the chemotherapy group.

Concerning PDT therapy, a large cluster of genes was also downregulated after treatment (Supplementary Figure 44), including the genes mainly involved in membrane tracking, protein folding, and signal transduction, such as ARL17A (ADP ribosylation factor like GTPase 17A) and VPS13B (vacuolar protein sorting 13 homolog B). The inhibition of genes related to cell proliferation, such as PCID2 (PCI domain containing 2), NHRNPU-AS1 (heterogeneous nuclear ribonucleoprotein U, scaffold attachment factor A), KIZ (kizuna centrosomal protein), and POLQ (DNA polymerase theta) were also clearly identified. The alternation of genes in immune response mechanisms, including CXCL14 (chemokine C-X-C motif ligand 14) and BCL2L11 (bcl-2-like protein 11) are also suggested.

After photochemotherapy (MNPs + L), the tumors showed the largest group of altered genes compared with the untreated group (Supplementary Figure 45). The cluster of upregulated genes contained those mainly involved in metabolism such as GGT1 (gamma glutamyl transferase 1), vesicular traffic/secretion such as SYT13 (synaptotagmin XIII) and tumor suppression such as ARMCX3 (armadillo repeat-containing X-linked protein 3) and ARMCX6. The cluster of downregulated genes contained those mainly involved in cytoskeleton organization such as COL5A2 (collagen V, alpha 2), cell–cell interaction such as EDIL3 (EGF like repeats and discoidin domains 3) or immune response mechanisms such as CXCL8 (chemokine C-X-C motif ligand 8).

Kyoto Encyclopedia of Genes and Genomes (KEGG) and supervised gene ontology (GO) pathway analyses were carried out to depict the biological processes and point out the molecular pathways. Gene Set Enrichment Analysis (GSEA) tools were used for pathway analyses on the basis of the genes expressed differentially in these therapeutic groups compared with the untreated control group. Differential expression was defined as multiple testing adjusted $p$-value <0.05 and fold change ≥1.5. Based on GSEA, the most enriched KEGG pathways, GO processes, and disease biomarkers were summarized (Supplementary Figures 46–48), confirming the distinct alternation pattern of genes in response to different therapeutic modalities. For chemotherapy alone, the molecular pathways involved were mainly adjusted by the genes regulating metabolism of nucleotides, amino acids, or carbohydrates (Supplementary Figure 46). The enriched pathways after PDT approach were mainly regulated by intracellular signaling and transcription (Supplementary Figure 47). For the group receiving photochemotherapy, the molecular pathways involved those mainly controlled by the genes related to cell–cell/ECM interaction and intracellular signaling (Supplementary Figure 48).

To recognize the key genes involved in tumor that were changed after different therapies, a Venn diagram was developed manually (Fig. 6e). The results showed that 9 key genes were significantly changed for the mice receiving the three different treatments (Fig. 6f). Among them, CDC-like kinase 4 (CLK4) gene encodes protein kinase that can interact with and phosphorylate the arginine- and serine-rich proteins, which play key roles in the formation of spliceosomes, and thus are perhaps involved in the regulation of alternative splicing. SRSF1 (serine and arginine rich splicing factor 1) gene encodes a member of the serine/arginine-rich splicing factor proteins. The encoded proteins either repress or activate splicing, depending on the interaction partners and phosphorylation state, playing an important role in regulating splicing. Besides, SNRNP70 (U1 small nuclear ribonucleoprotein 70 kDa) associated with U1 spliceosomal RNA that assists the formation of the spliceosome was also downregulated. Therefore, downregulation of the CLK4, SRSF1, and SNRNP70 genes may suppress cell growth and induce apoptosis[53–55]. Furthermore, western blot analysis was performed to validate the expression level of these 9 key proteins (Supplementary Figure 49), which was in good agreement with the gene expression results, further confirming the accuracy and reliability of the gene chip approach.

**Combating drug-resistant cancer.** Apart from systemic toxicity, the anticancer efficacy of platinum-based drugs is often limited by the inherent or acquired resistance possessed by various cancers[56]. Great efforts have been made to overcome drug resistance, but only limited achievements have been realized in clinical applications[57,58]. For example, high dose regimens showed slightly better efficacy in some cases but were always accompanied with enhanced systemic toxicity. Moreover, new chemotherapeutic anticancer drugs or their combination with chemosensitizing agents remain discouraging in clinical trials. Development of new treatment strategies is urgently needed in order to eradicate cancer more effectively. PDT is a perfect partner for chemotherapy in combating drug resistance through different anticancer mechanisms to achieve enhanced anticancer efficacy. Encouraged by the excellent synergistic efficacy of the MNPs on U87MG tumor model, we tested the anti-tumor efficacy of the MNPs on a cisplatin-resistant tumor model.

Compared with other formulations, MTT assay indicated that the cytotoxicity of MNPs upon irradiation against drug-resistant A2780CIS cell was greatly improved combining chemotherapy and PDT (Fig. 7a and Supplementary Figure 50), demonstrating their excellent anticancer activity. The synergistic anticancer efficacy was further confirmed by Annexin V-FITC/PI assay (Fig. 7b), where much higher later apoptosis and necrosis were detected for the cells after photochemotherapeutic treatment than those of chemotherapy or PDT alone. The mice bearing A2780CIS tumors were administered with a single dose of PBS, light, cPt, cisplatin, MNPs, TPPNPs + L, and MNPs + L. Clearly, mono-chemotherapeutic treatments (cisplatin, cPt, and MNPs) were ineffective in suppressing tumor growth (Fig. 7c, e). Although single PDT formulation outperformed the mono-chemotherapeutic treatments in reducing tumor volume, tumor recurrence still occurred at 9 days post treatment. In sharp contrast, the combination of chemotherapy and PDT (MNPs + L) effectively ablated all the tumors without recurrence during the course of the therapy. PDT eliminated the primary tumor tissue through local irradiation and chemotherapeutic drug killed the residual cancer cells, thus effectively inhibiting tumor recurrence. The median survival time was determined to be 33, 35, 40, 43, 53, and 62 days for the mice treated with PBS, light, cPt, cisplatin, MNPs, and TPPNPs + L, respectively (Fig. 7f). Notably, photochemotherapy administration greatly prolonged mice survival over 90 days without a single death. Other studies

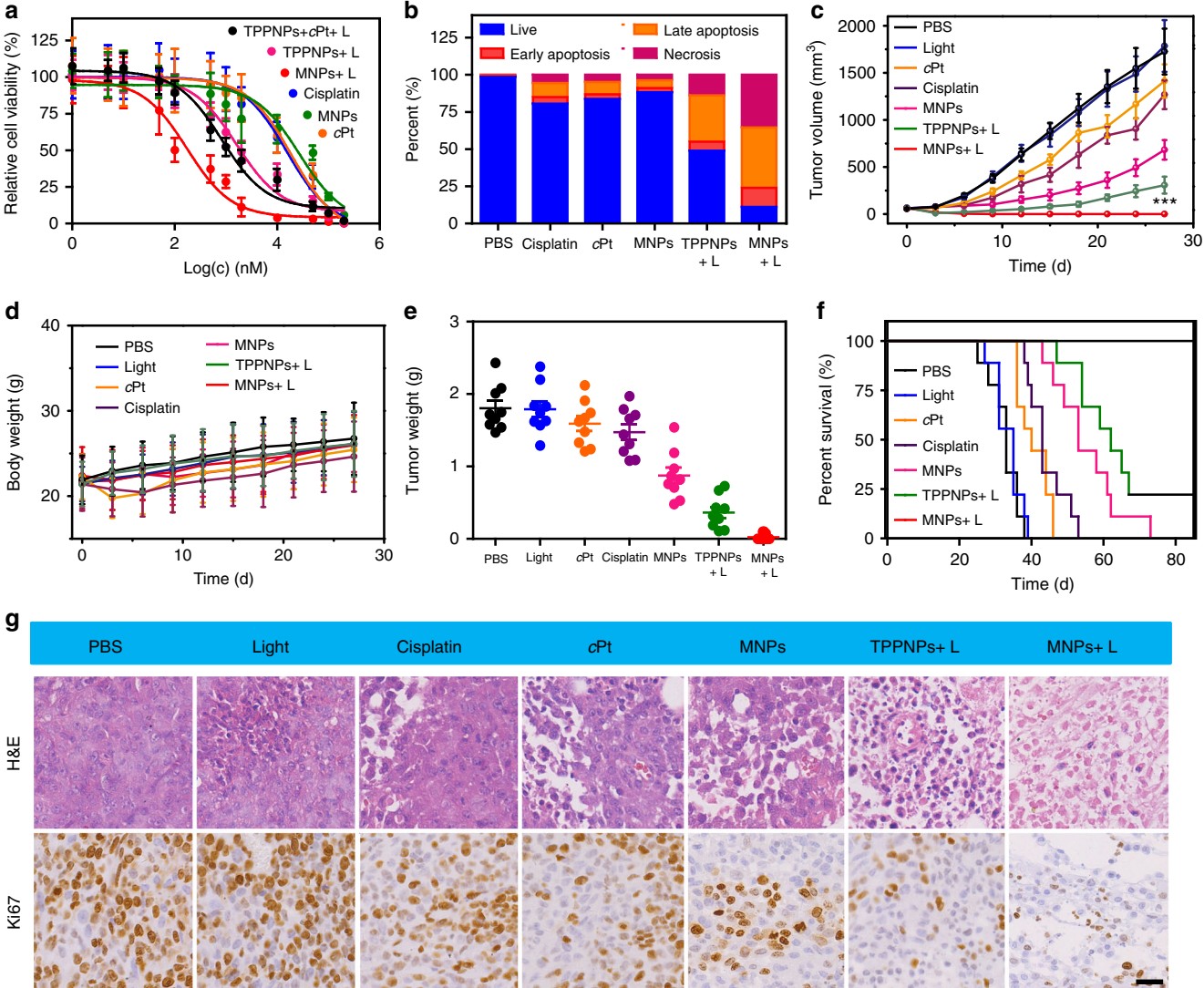

**Fig. 7** In vivo anti-tumor efficacy against resistant cancer model. **a** Cytotoxicity of A2780CIS cells treated with different administrations. The irradiation density was 0.1 W cm⁻² at 671 nm and the irradiation time was 3 min. **b** Annexin V/PI analyses of A2780CIS cells after different treatments. **c** In vivo tumor growth inhibition curves and **d** body weight changes of the mice after different treatments (*n* = 9). **e** Weight of A2780CIS tumors after different treatments. **f** Kaplan–Meier plots of the mice treated with different treatments. The irradiation density was 0.3 W cm⁻² at 671 nm and the irradiation time was 10 min. **g** H&E and Ki67 analyses of tumor tissues after different treatments. Scale bar is 50 μM. The data are expressed as means ± s.d., ***P < 0.001

including body weight assessment (Fig. 7d) and immunohisto-chemical analyses (Fig. 7g) all supported the notion that photochemotherapy was capable of thoroughly inhibiting tumor growth on the cisplatin-resistant A2780CIS tumor model with negligible systemic toxicity.

**Combating orthotopic cancers and inhibiting cancer metastasis.** In the clinic, cancer metastasis suffers from poor prognosis, leading to ~90% of cancer-related mortalities[59,60]. For example, metastatic breast cancer is largely incurable among women patients and the 5-year survival rate is only around 20% caused by its high metastasis in lung, liver, and bones. Although great efforts have been made over the past 30 years, the therapeutic outcomes are still poor for most chemotherapies and the overall survival is improved by only a few months at most[61–63]. Inspired by the excellent anticancer efficacy against U87MG and A2780CIS models, the combination of chemotherapy and PDT will be an ideal choice to treat highly aggressive 4T1 breast cancer, because

the cancer cells in the primary tumor site can be completely eliminated, preventing the escape of the surviving cancer cells to new organs. Indeed, MTT assay confirmed that the photo-chemotherapy (MNPs + L) exhibited superior synergistic antic-ancer efficacy against 4T1 cells (Supplementary Figure 51).

In order to verify the in vivo anti-tumor performance and anti-metastasis effect of photochemotherapy, 4T1 tumors were orthotopically inoculated in the mammary fat pads to produce spontaneous metastases in the lung, which was an experimental animal model for stage IV human breast cancer. 4T1 tumor-bearing mice were randomly divided into six groups (*n* = 8/group) and formulated with PBS, light, *c*Pt, MNPs, TPPNPs + L, or MNPs + L, respectively. As shown in Fig. 8a, the tumors from the mice treated with PBS and light grew exponentially. Compared with these two control groups, very limited anti-tumor efficacy was monitored for the mice treated with *c*Pt after a single-dose injection. Meantime, moderate tumor inhibition was monitored for the mice formulated with MNPs benefiting from EPR effect and active targeting capability. A decrease in tumor

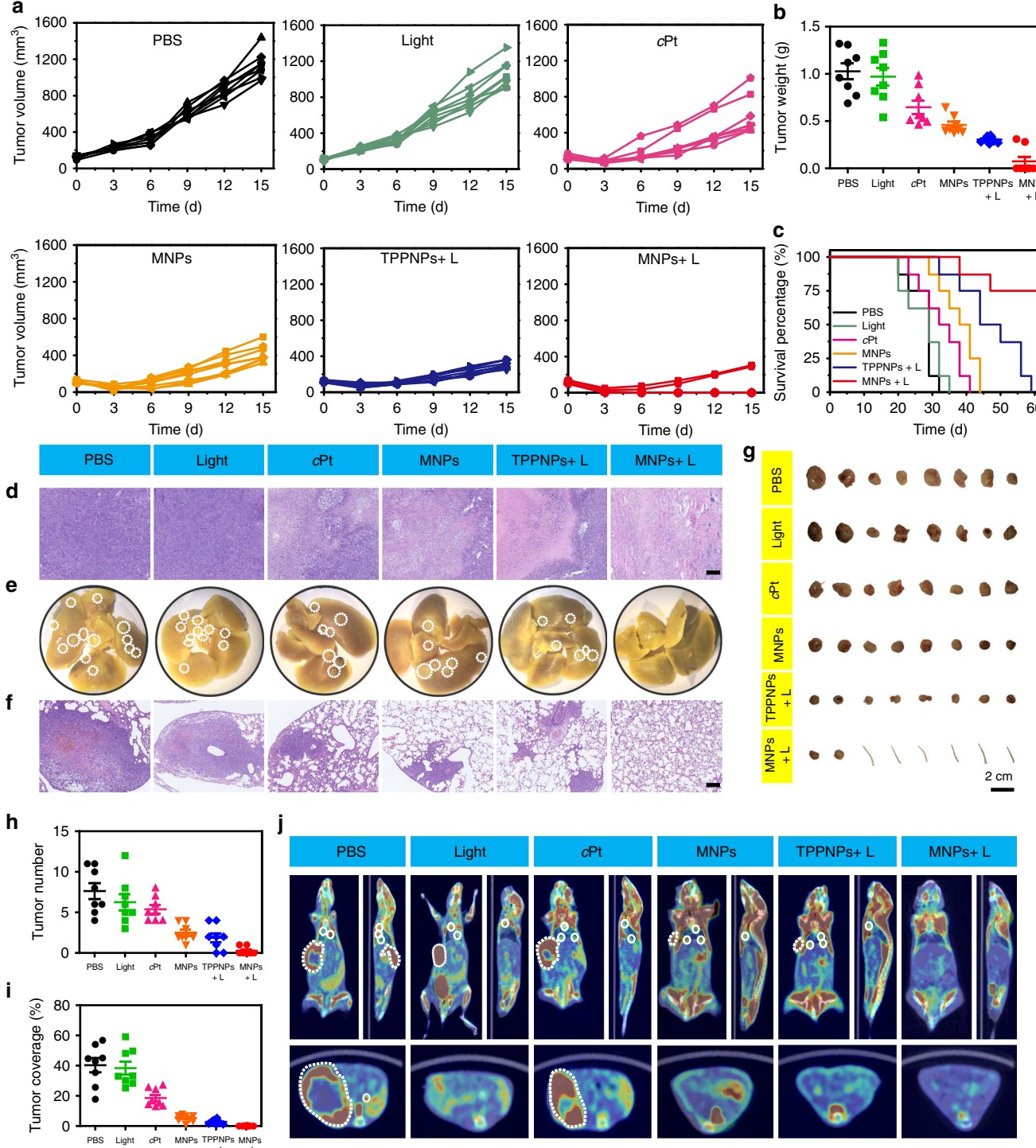

**Fig. 8** Treatment of orthotopic breast cancer and inhibitory effects on lung metastasis. **a** Tumor volume changes of the mice treated with PBS, light, cPt, MNPs, TPPNPs + L, or MNPs + L after one injection (n = 8). **b** Tumor weight of the mice treated with different formulations. **c** Kaplan–Meier survival curves of the mice bearing orthotopic 4T1 breast tumors treated with different formulations. **d** H&E staining of the tumor tissues from each group. Scale bar is 500 µM. **e** Representative images of the lungs excised from each group. The white circles denote the visually detected metastatic nodules in each lung tissue. **f** Histological examination of metastatic lesions in lung tissues from each group after H&E staining. Scale bar is 500 µM. **g** Photo images of the orthotopic tumors harvested from the mice treated with different formulations. Mice tails indicate no tumor founded. **h** The numbers of tumor nodules present in the lungs from each group. **i** Tumor coverage percentage in the lungs from each group. **j** PET/CT images of the mice treated with different formulations at the 14th day post injection. The white dash circles denote the orthotopic breast tumors, and the white solid circles denote the metastatic tumors. The laser density was 0.3 W cm$^{-2}$ at 671 nm, and the irradiation time was 10 min. The data are expressed as means ± s.d., ***$P < 0.001$

volume was monitored, while recurrence occurred after 6 days post injection, confirming that single chemotherapy hardly suppressed the tumor growth. Compared to the PBS-treated group, better anti-tumor effect was achieved for the mice receiving PDT (TPPNPs + L) with a 68.1% reduction in tumor volume. However, tumor recurrence still occurred after 6 days post injection, because PDT alone did not completely eliminate all cancer cells in tumor sites. Notably, the combination of chemotherapy and PDT exhibited superior anti-tumor efficacy with a 93.5% reduction in tumor volume. Attributing to the synergistic anticancer efficacy, the primary 4T1 tumors in six mice were completely eradicated without any recurrence during the therapeutic period (Supplementary Figure 52, Fig. 8a). The mice were killed and the excised tumors were photographed and weighed (Fig. 8g). The average tumor weight was 1.02, 0.97, 0.65, 0.46, 0.30, and 0.07 g for the mice treated with PBS, light, $c$Pt, MNPs, TPPNPs + L, or MNPs + L, respectively (Fig. 8b), in agreement with the results in Fig. 8a. Analysis of the H&E stained tumor sections confirmed considerably enhanced necrosis in the photochemotherapy group (~95% necrotic area) when compared with the PBS-treated group (~10% necrotic area), indicating the combination of chemotherapy and PDT was the most effective in suppressing tumor growth (Fig. 8d). Additionally, no weight loss was detected for the mice injected with MNPs followed by irradiation, suggesting low systemic toxicity of the photochemotherapy (Supplementary Figure 53). Due to the excellent anti-tumor performance and negligible systemic toxicity of photochemotherapy, the median survival was remarkably increased (Fig. 8c), in sharp contrast with the mice receiving chemotherapy ($c$Pt and MNPs) or PDT alone.

At the end of the study, mice were killed and the lung tissues from each group were carefully collected and photographed (Supplementary Figures 54−59 and Fig. 8e). The number of metastatic tumor nodules and tumor coverage percentage were recorded to assess the anti-metastasis effect of these formulations. The average metastatic tumor nodules per lung were counted to be 7.63, 6.35, 5.88, 2.50, and 1.88 for the mice treated with PBS, light, $c$Pt, MNPs, and TPPNPs + light, respectively (Fig. 8h). According to the H&E staining of the lung tissue sections, the tumor coverage percentage was calculated to be 40.3, 38.4, 18.5, 5.3, and 2.8% for the mice treated with PBS, light, $c$Pt, MNPs, and TPPNPs + light, respectively (Fig. 8f, i), demonstrating that only slight or moderate anti-metastasis effect was achieved by chemotherapy or PDT alone. Excitingly, only one tumor nodule was visualized in two mice administrated with photochemotherapy, the average tumor nodules were 0.25 and only 0.1% of lungs were occupied by tumors, indicating superior anti-metastatic efficacy of photochemotherapy. By using $^{18}$F-fluorodeoxyglucose (FDG) as a radiotracer, PET/CT imaging was further employed to detect metastatic lung tumors. Compared with other groups, barely any metastatic nodules were observed from the randomly selected mice treated with photochemotherapy (Fig. 8j). These investigations firmly demonstrated that the combination of chemotherapy and PDT not only successfully suppressed primary tumor growth but also effectively inhibited lung metastasis.

Moreover, an orthotopic hepatoma model was established to validate the in vivo anti-tumor activity of photochemotherapy and its possible damage to normal liver. The orthotopic hepatoma model was generated by surgical implantation of a small tumor section about 1 mm$^3$ into the liver. Once tumors formed (~100 mm$^3$), the hepatoma-bearing mice were randomly divided into five groups and treated with PBS, $c$Pt, MNPs, TPPNPs + L, or MNPs + L, respectively. The therapeutic efficacies of these administrations were determined by measuring tumor sizes as estimated from MRI images and physical examination of the excised livers. As shown in Figure 9a–e, the anti-tumor efficacy

was unsatisfactory for the mice receiving chemotherapy or PDT alone after a single injection. Compared with the control group, only limited anti-tumor results were monitored in the mono-therapy groups ($c$Pt, MNPs, and TPPNPs + L), where the tumors grew gradually. The mice receiving MNPs + L demonstrated significant tumor inhibition, the tumor size decreased after photochemotherapy without recurrence during the experimental period.

Monitored by MRI, the tumor was almost annihilated after photochemotherapy, only a small spot was observed in the liver possibly corresponding to necrotic tumor tissue (Fig. 9j). However, the tumor grew fast in the other four groups, suggesting that these administrations failed to treat orthotopic hepatoma. At 21 days after treatment, all tumors were collected with their surrounding liver tissue (Fig. 9g). Different from the other four groups exhibiting heavy tumor burden in liver tissues, the tumor volumes were remarkably smaller for the mice treated with photochemotherapy. It should be emphasized that three tumors were completely ablated from livers, benefiting from the synergistic efficacy. The therapeutic performances were further accessed via H&E and Ki67 staining of the tumor tissues collected at end-point necropsy, providing convincing evidences for the apoptosis degree of tumor tissues. Photochemotherapy resulted in the largest region of necrosis and the least proliferation in the tumor tissue (see white dotted lined region in the images), but no apparent apoptosis and damage were detected in the normal liver tissue (Fig. 9h). PET/CT imaging also confirmed that no tumor was detectable in the mice administered with photochemotherapy (Fig. 9i), confirming the excellent synergistic efficacy far exceeded the outcomes from mono chemotherapy or mono PDT.

The systemic toxicity of these formulations was also evaluated by recording the body weight changes during the treatments. A slight reduction in body weight was monitored for the mice receiving photochemotherapy in the first three days because of surgical trauma, then the body weight gradually increased, suggesting that no noticeable systemic toxicity occurred in vivo (Supplementary Figure 60). Moreover, no obvious signs of toxic effects, such as changes in drinking, eating, grooming, activity, urination, or neurological status took place during the treatment period. However, the body weight of the mice in the other groups continuously decreased caused by the heavy tumor burden in the liver. In addition, the physical health conditions of mice in these four groups was extremely poor, the mice displayed lethargy during the observation period. From the Kaplan–Meier survival curves (Fig. 9f), the median survival rate was determined to be 39, 40.5, 48, and 52.5 days for the mice administrated with PBS, $c$Pt, MNPs, and TPPNPs + light, respectively. Excitingly, the group administrated with MNPs + L had a 100% survival rate, which reflected the superior anti-tumor efficacy and good quality of life in these mice. These studies suggested that even though the tumors were inoculated located in livers, the photochemotherapy was able to precisely and effectively kill the tumor cells, demonstrating that our nanomedicine possessed promising application in the treatment of hepatoma.

## Discussion

We prepared a sophisticated porphyrin-based metallacage through multicomponent coordination-driven self-assembly, acting as a theranostic platform to fabricate MNPs. Attributing to the nano-formulation by two special amphiphilic diblock polymers (mPEG-$b$-PEBP and RGD-PEG-$b$-PEBP), the resultant MNPs exhibited long blood circulation time and high tumor accumulation benefiting from EPR effect and active targeting ability, playing an important role in improving anticancer efficacy and reducing side effect toward normal tissues. The fluorescence

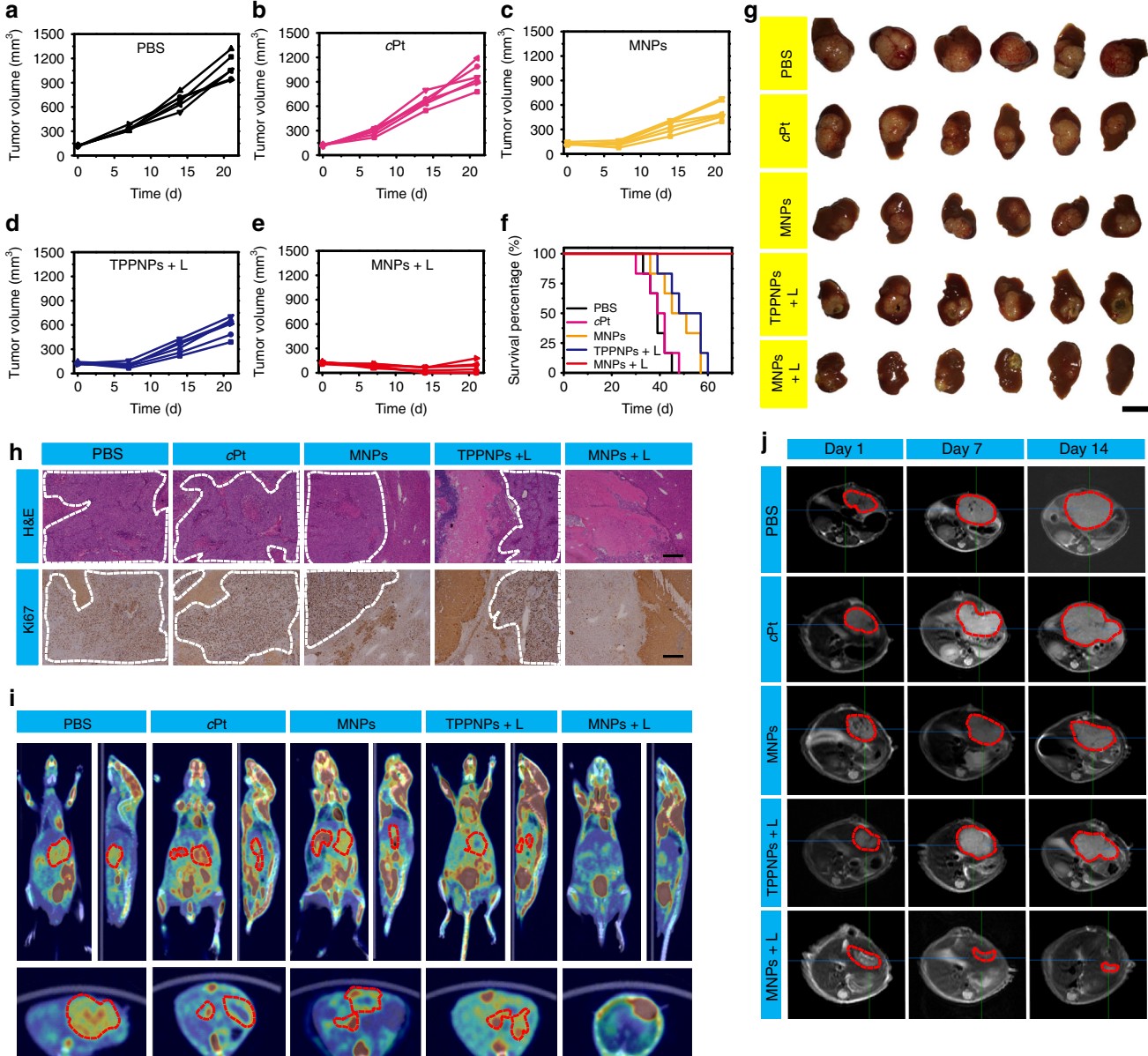

**Fig. 9** Treatment of orthotopic liver cancer. Tumor volume changes of the mice bearing orthotopic LM3 tumors treated with **a** PBS, **b** $c$Pt, **c** MNPs, **d** TPPNPs + L, or **e** MNPs + L after one injection ($n = 6$). **f** Kaplan–Meier survival curves of the mice bearing orthotopic LM3 tumors treated with different formulations. **g** Photo images of the liver tissues containing LM3 tumors collected from the mice treated with different formulations. **h** H&E and Ki67 staining of the liver tissues from each group. The white dashed circles denote the tumor tissues. Scale bar is 500 μM. **i** PET/CT images of the mice treated with different formulations. The red dash circles denote the orthotopic hepatocellular carcinoma tumors. (**j**) MRI images of the mice treated with different formulations. The red dashed circles denote the orthotopic hepatocellular carcinoma tumors. The laser density was 0.3 W cm$^{-2}$ at 671 nm, and the irradiation time was 10 min. The data are expressed as means ± s.d

emission and $^1O_2$ generation quantum yield of the porphyrins were dramatically increased upon formation of MNPs, which was favorable for NIRFI and PDT. By chelating a positron emitting metal ion ($^{64}$Cu) or a paramagnetic Mn, the $^{64}$Cu@MNPs (or Mn@MNPs) are excellent PET imaging and MRI agents, allowing precise diagnosis of tumor and real-time monitoring of delivery, biodistribution and excretion of the MNPs. The combination of chemotherapy and PDT displayed superior synergistic efficacy in vitro and in vivo. Indeed, superior tumor suppression was realized without recurrence after single-dose injection against U87MG, cisplatin-resistant A2780CIS and orthotopic (4T1 and LM3) tumor models, and excellent anti-metastasis effect was

achieved attributing to the synergistic photochemotherapy. This study provides a picturesque strategy to develop multifunctional theranostic systems, providing a blueprint for the next generation of nanomedicines.

## Methods

**Preparation of M**. $c$Pt (29.2 mg, 40.0 mM), TPP (6.18 mg, 10.0 mM), and DSTP (4.20 mg, 20.0 mM) were placed in a 2-dram vial, followed by addition of acetone-$d_6$ (1.6 mL) and D$_2$O (0.4 mL). After stirring at 70 °C for 5 h, the solvent was removed by N$_2$ flow. Acetone-$d_6$ (0.6 mL) was added, and the solution was stirred at 70 °C for an additional 5 h. The formed M was precipitated using diethyl ether and the precipitation procedure was repeated four times to purify M (yield, 91.6%).

**Preparation of MNPs**. For in vitro and in vivo studies, the MNPs were fabricated through a matrix-encapsulation method. mPEG-*b*-PEBP (25.0 mg), RGD-PEG-*b*-PEBP (5.00 mg), M (10.0 mg) and were solubilized in 5 mL of acetone, then the solution was injected into MilliQ water (20 mL) with sonication of the mixture for 5 min. After sonication for 5 min, the organic solvent was completely removed in vacuo. After sonication for another 5 min, a well dispersed nanoparticle suspension was obtained. Unloaded M in the solution was eliminated by passing through a PD10 column. DLS and TEM were used to characterize these MNPs. UV-vis spectroscopy was employed to confirm the actual loading amount of M according to a standard curve. The drug loading content (LC) was determined using the following formula:

$$LC = m_M/(m_M + m) \times 100\%,$$

where $m_M$ represents the mass of M encapsulated in MNPs, and $m$ represents the mass of the copolymers (mPEG-*b*-PEBP and RGD-PEG-*b*-PEBP) used for the fabrication of MNPs.

**Detection of Production of $^1O_2$ in Solution**. In a typical experiment, MNPs (the concentration of TPP was 500 nM) were suspended in aqueous solution containing 5.00 μM of singlet oxygen sensor green (SOSG) dye, which is a singlet oxygen inductor. The mixture was then placed in a cuvette and the solution was irradiated at 671 nm (0.5 W cm$^{-2}$) for different time, with the fluorescence emission of SOSG (upon excitation at 470 nm) measured using a fluorescence spectrophotometer. NaN$_3$ was used as a $^1O_2$ quencher to further confirm the formation of $^1O_2$ by MNPs upon laser irradiation.

**MRI phantom study**. Various metal concentrations of Mn@MNPs or DTPA-Gd were prepared for MRI phantom study, ranging of 0.063, 0.125, 0.25, 0.50, and 1.00 mM for Mn or Gd ions. The longitudinal relaxation times were measured at 298 K on a Bruker 7T scanner (Pharmascan) for the calculation of the relaxation rate of the samples. The MR images were collected by using a spin echo sequence with parameters as follows: repetition time = 6000, 4000, 2000, 1000, 500, 250, 100 ms, echo time = 10 ms, matrix = 256 × 256, field of view = 40 × 40 mm$^2$, slice thickness = 3.00 mm.

**Cell culture**. U87MG, A2780, 4T1, and LM3 cell lines were purchased from American Type Culture Collection (ATCC, Rockville MD), A2780CIS cell line was purchased from Sigma. U87MG cells were incubated in Minimum Essential Medium (MEM) containing FBS (10%) and penicillin/streptomycin (1%). 4T1, A2780, and LM3 cells were cultured in RPMI-1640 medium containing FBS (10%) and penicillin/streptomycin (1%). LM3 cells were incubated in Dulbecco's Modified Eagle's Medium (DMEM) containing FBS (10%) and penicillin/streptomycin (1%). A2780CIS cells were incubated in cisplatin-containing medium (2 μg mL$^{-1}$) for 10 days before experiments.

**Cytotoxicity evaluation**. The cytotoxicity against the cell lines was evaluated by MTT assay. The cells were seeded at a density of $1.00 \times 10^4$ cells/well, and cultured for 24 h for attachment. Then the cells were cultured with fresh serum-supplemented medium without/with cisplatin, *c*Pt, TPPNPs, TPPNPs + *c*Pt (TPPN/*c*Pt = 1:4, molar ratio), and MNPs at various concentrations for 24 h. For the groups treated with TPPNPs (mono-PDT) and MNPs (photochemotherapy), the cells were irradiated with light at 671 nm (0.1 W cm$^{-2}$, 3 min) after 12 h incubation. After irradiation, the cells were further incubated for 12 h. Then MTT solution (20 μL, 5.00 mg mL$^{-1}$) was added to each well. After incubating the cells at 37 °C for 4 h, the MTT solution was removed and the cells were washed three times by PBS. DMSO (100 μL) was added to solubilize the insoluble formazan crystals, and the absorbance was measured by a spectrophotometer (570 nm). The cells without any treatment were utilized as a control. To confirm internalization via receptor-mediated endocytosis, a competition assay was performed that involved pre-treating with free cRGDfK (20 μM) for 30 min. The culture media were refreshed and treated with the MNPs followed by irradiation after 12 h culture and analyzed by MTT assay after another 12 h culture. All experiments were carried out with five replicates.

**In vivo tri-modality imaging**. When the tumor size reached around 100 mm$^3$, mice were i.v. injected with MNPs. In vivo fluorescence imaging was performed on an IVIS Kinetic Imaging System with excitation filter of 570 nm and emission filter of 600–800 nm.

For PET imaging, $^{64}Cu$ MNPs (150 μCi) were intravenously injected into the U87MG tumor-bearing mice. PET imaging and data analysis were performed on an Inveon microPET scanner (Siemens Medical Solutions). 3D ROI was drawn over the tumor and organs on decay-corrected whole-body coronal images. The average radioactivity concentration was obtained from the mean pixel values within the ROI volume, which was converted to counts per milliliter per minute by using a predetermined conversion factor. At 48 h post injection, the mice were killed, and the radioactivity in the organs (heart, liver, stomach, lung, kidney, brain, spleen, intestine, bladder, pancreas, bone, muscle, and tumor) was measured on a well Beckman 8000 gamma counter (Beckman, Brea, CA).

For MRI, the mice were treated with an intravenous injection of Mn@MNPs (10 mg kg$^{-1}$) or Magnevist (5.0 mg kg$^{-1}$). MRI was recorded on a high magnetic field micro-MR scanner (7.0T, Bruker, Pharmascan). $T_1$-weighted images were collected by a (MSME) sequence and the parameters were as the following: repetition time (TR), 600 ms; echo time (TE), 24 ms; flip angle = 180; number of excitations (NEX), 1; matrix size, 256 × 256; field of view (FOV), 4 × 4 cm$^2$; slice number, 16; slice thickness, 1 mm. Signal intensities were measured in defined ROI with Image J software. The SNR values were calculated using the formula SNR = $S_I/S_D$ ($S_I$ and $S_D$ were signal intensity and standard deviation, respectively).

**Tumor model**. BALB/c nude mice were purchased from Zhejiang Academy of Medical Sciences and maintained in a pathogen-free environment under controlled temperature. Animal care and handing procedures were in agreement with the guidelines evaluated and approved by the ethics committee of Zhejiang University. Study protocols involving animals were approved by the Zhejiang University Animal Care and Use Committee. The nude mice were subcutaneously injected with U87MG (or A2780CIS) cell suspension (200 μL, $5 \times 10^6$) in the right flank region. The tumors were allowed to grow to ~100 mm$^3$ before experimentation. The tumor volume was calculated as (length) × (width)$^2$/2.

**Pharmacokinetics and tissue distributions**. *c*Pt or MNPs was i.v. injected into the mice at a dose of 2 mg Pt per kg. Blood was collected ($n = 4$ for each group) at 15 min, 1 h, 2 h, 4 h, 8 h, 12 h, 24 h, and 48 h post injection and kept in heparinized tubes. The amount of platinum in the plasma was determined by ICP-MS. For biodistribution analysis, the main organs (liver, lung, spleen, kidney, stomach, brain, intestine, bladder, and tumor) were excised at 2, 6, 12, 24, and 48 h post injection and kept in dry ice. Organs were digested using concentrated nitric acid and the amount of platinum was analyzed by ICP-MS.

**In vivo anti-tumor activity**. The mice were randomly divided into seven treatment groups ($n = 9$) when the mean tumor volume (U87MG and A2780CIS tumor models) reached around 100 mm$^3$ and this day was set as day 0. Mice were received different formulations: PBS, light, *c*Pt, cisplatin, MNPs, TPPNPs + L, and MNPs + L. The dosage of Pt and porphyrin (TPP or TPPN) was 2.00 mg Pt kg$^{-1}$ (10.3 μM Pt kg$^{-1}$) and 2.58 μM kg$^{-1}$ in these formulations, respectively. Since the lower laser power density with shorter irradiation time is much desired in the invasive tumor therapy, the optimized 0.3 W cm$^{-2}$ power density of 671 nm laser with 10 min duration was chosen herein for PDT (TPPNPs + L) and photochemotherapy (MNPs + L) at 12 h post i.v. injection. Tumor volume and body weight were measured every 3 days. The tumors were excised at day 36 post treatment, and the weight of the tumor was assessed.

**Gene expression analysis of tumors**. The gene expression profiles were tested by using an Affymetrix GeneChip PrimeView™ human gene expression array, which contained more than 520,000 probes housing over 36,000 transcripts and variants, including over 20,000 well annotated Refseq genes. A cluster of genes that are differentially expressed in tumors receiving chemotherapy, PDT and photo-chemotherapy were identified. From the unsupervised hierarchical clustering analyses, different gene expression profiles clearly exhibited for the mice treated with different therapies. A total of 19,985 genes were analyzed and an absolute fold change ≥1.5 compared with the untreated mice with an adjusted *p*-value <0.05 was used to define the differentially expressed gene sets.

The high-throughput gene analysis allowed us to classify and identify numerous core genes and molecular pathways determining the utility of anti-tumor response, which were previously poorly understood. In comparison with chemotherapy or PDT, the whole-gene expression analyses of tumors suggested that photochemotherapy induced the upregulation and downregulation of a much higher number of genes, which were responsible for its greater efficacy in eradicating tumor and improving mice survival. Moreover, molecular pathways controlling intracellular signaling and cell–cell/ECM interaction were 'turned on' following photochemotherapy, whereas metabolism and transcription mechanisms were stimulated in response to chemotherapy or PDT.

**Treatment of orthotopic breast cancer and inhibitory effects on lung metastasis**. To confirm the anticancer efficacy against orthotopic breast cancer and anti-metastasis effect in vivo, 4T1 cells ($5 \times 10^5$) suspended in 0.1 mL of saline were xenografted into the mammary fat pad of the mice. Anti-tumor treatments were performed when the tumors were around 130 mm$^3$. The mice were randomly divided into six groups ($n = 8$) and treated with PBS, light, *c*Pt, MNPs, TPPNPs + L, MNPs + L. The dosage of Pt and porphyrin (TPP or TPPN) was 2.00 mg Pt kg$^{-1}$ (10.3 μM Pt kg$^{-1}$) and 2.58 μM kg$^{-1}$ in these formulations. The irradiation density was 0.3 W cm$^{-2}$ at 671 nm, and the irradiation time was 10 min. Tumor volumes and mice body weight were measured every 3 days after the treatment using a caliper and an electronic balance. Fourteen days after treatment, mice were injected with $^{18}F$-FDG (~0.2 mCi). PET/CT imaging was conducted on a microPET-CT scanner (Siemens Preclinical Solution USA, Inc., Knoxville, TN, USA) to detect the metastatic tumors in lung tissues after different treatments. The mice were killed at the end time point, the tumor and lung tissues from each group were carefully removed and photographed. The number of metastatic nodules from each group were counted and recorded to evaluate the inhibition of lung metastasis. Moreover,

the lung tissues from each group were assessed by histological examinations to detect the metastatic lesions.

**Treatment of orthotopic hepatoma**. BALB/c nude mice (around 5 weeks) were used in the experiments. The orthotopic hepatoma model was generated by surgical implantation of a small tumor tissue about 1 mm$^3$ into the liver. The tumor size was followed by MRI (GE Signa HDxT, 3.0 T). Anti-tumor treatments were performed when the tumors were around 100 mm$^3$. The mice were randomly divided into five groups ($n = 6$) and treated with PBS, $c$Pt, MNPs, TPPNPs + L, or MNPs + L, respectively. The dosage of Pt and porphyrin (TPP or TPPN) was 2.00 mg Pt kg$^{-1}$ (10.3 μM Pt kg$^{-1}$) and 2.58 μM kg$^{-1}$ in these formulations. 12 h post injection, the animal was anaesthetized and the liver was exposed for irradiation. The irradiation density was 0.3 W cm$^{-2}$ at 671 nm, and the irradiation time was 10 min. Fourteen days after treatment, PET/CT imaging was conducted to monitor the tumors in liver tissues after different treatments. The mice were killed at the end time point, and the tumor and liver tissues from each group were carefully removed, photographed, and further examined by histological examination.

**Other methods**. Other information about syntheses, characterizations, in vitro studies, and in vivo investigations are given in the Supplementary Information.

## Data availability

All data are available from the authors upon reasonable request.

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

## Acknowledgements

This work was supported by the Intramural Research Program of the National Institute of Biomedical Imaging and Bioengineering, and National Institutes of Health, National Key R&D Program of China (2016YFB0700804), National Natural Science Foundation of China (51673171, 21620102006), and Zhejiang Provincial Natural Science Foundation of China (Grant LR16E030001).

## Author contributions

G.Y., Z.M., F.H., P.J.S., and X.C. conceived and designed the research. G.Y., J.Z., and L.S. prepared the metallacage and measured the photophysical properties. Y.L. and Z.Z. performed the PET imaging and MRI studies. G.Y., Y.L., F.Z., and S.W. performed the in vitro experiments and analysed the data. G.Y., S.Y., J.C., C.G., and Z.M. performed the in vivo experiments and analysed the data. G.Y., M.L.S., T.R.C., B.C.Y., Z.M., F.H., P.J.S., and X.C. co-wrote the paper.

## Additional information

**Competing interests:** The authors declare no competing interests.

