## [Peer Review File · Nature Communications]

Reviewers' Comments:

Reviewer #1:

Remarks to the Author:

In this manuscript, the authors prepared organoplatinum(II) metallacage (M) loaded nanoparticles through a self-assembly process, which showed much improved singlet oxygen quantum yield and great promise for multimodal imaging and treatment synergy. Overall, the hypothesis is sufficiently supported by experiment data and the work is significant and should be appealing to the broad readership. Some suggestions are listed below to improve this manuscript further.

Some comments:

1. Please briefly comment on the advantages and novelty of this particular system over other phototheranostic nanoparticles.
2. Please revise the schematic diagrams for improved simplicity and clarity for readers. For example, in Figure 1 (a), labels should be added for every structure. In Figure 1(b), the logical relationship amongst the three components could be made clearer.
3. More detailed analysis should be added from figures S4 to S9 to demonstrate that the structures were successfully synthesized. In the captions, full names for the abbreviations should be given. For example, what does TPPN stand for?
4. Detailed information should be described for the loading content calculation by UV. The content percent of every element in EDS study should be added.
5. Please provide further evidence to demonstrate the serum stability of MNPs. Suggest to increase the percent of serum from 10% to 50% FBS for particle stability assessment.
6. Please add results of 4h post injection in the text corresponding to Fig. 4e, etc. The authors should check their description on Page 15, line 7 and line 12 and Page 34, line 18. Thorough proofread is needed.
7. Need strong references to support the claim that the half-life time of 1.87 ± 0.3 h is excellent. In Figure. 5(a), full pK analysis is needed (e.g., R2, t1, t2, etc.).
8. Please explain why "0.3 W cm⁻² power density of 671 nm laser with 10 min duration" were the optimal parameters?
9. It is important to investigate the drug release behavior from the nanoparticles in serum containing media or in different pH condition.
10. Ref 14 does not provide the right context to introduce PDT nor it is the first self-assembled nanoporphyrin agents. Suggest to either replace it with Nature Material 2011, 10, 324 or with another classic PDT agent review (Chem Rev 2010, 110, 2839-2857).

Reviewer #2:

Remarks to the Author:

The manuscript “A discrete organoplatinum(II) metallacage as a multimodality theranostic platform for cancer photochemotherapy” describes a new type of metallacage-loaded nanoparticles with improved photochemical properties for both multi-modal imaging and anti-cancer photodynamic therapy. The work is a large step forward in the use of self-assemble metallacage for therapeutic application based on the authors’ previous PNAS work (ref # 21). Most studies are well designed and performed in the manuscript. I highly recommend the acceptance of the manuscript if the authors can address questions below:

1. P5 “**M** was quantitatively obtained by stirring the mixture in acetone/water ($v/v = 8:2$) at 70 °C for 5 h. The well-defined signals in ^1H NMR spectra support the formation of a discrete and highly symmetric assembly as a sole thermodynamic product (Supplementary Fig. S10)²². ” In SI Fig. S10 the NMR was measured in CD_3CN , but in the synthesis procedure in SI PS3 “Then **M** was dissolved in CD_2Cl_2 for characterization.” Please verify the NMR solvent for Fig. S10. Also please make comparison of **M** with the mixture of TPP and DSTP in ^1H NMR spectra as the authors did in ref.22-23. Current Fig. S10 is less informative to show the distinctive formation of **M**.
2. The authors may provide the yield information about **M**, and whether or not a separation / purification step is needed or not. The authors may also consider to provide information about the scale –up capabilities of **M** and MNP, which could be advantageous for future translational studies.
3. “Metallocage” or “Metallacage” ? (both terms were seen in the literature)
4. P10 “A control experiment used **TPPNPs** fabricated from **TPPN**, **mPEG-*b*-PEBP** and **RGD-PEG-*b*-PEBP** with a loading content of 28.3%.” Please give the full name of **TPPN** and note it is different from TPP (the reader may think the TPPNP is a particle based on TPP instead of TPPN). It is also necessary to give the chemical design justification about TPPN (at least in SI). As the metallocage **M** is based on TPP and DSTP (benzene ring), why using naphthalene ring for TPPN synthesis instead of benzyl bromide? In addition, as TPPNPs were used for in vivo studies in the manuscript, it would be better to mention the use of TPPNP for photodynamic therapy as a control regimen at the first place.
5. P17 “...in order to extrapolate their pharmacokinetics (pK_a).” It is unusual that “pharmacokinetics” is abbreviated to “ pK_a ” (often referred to acid disassociation constant). Please revise the abbreviation.
6. P20 Figure 5a “ $n=3$.” At least n should be 4, and please compare the two groups statistically. Additionally, Figure 5b the group MNP is with light irradiation or not? Please compare intratumoral Pt accumulation for MNPs with and without light irradiation

to see whether photodynamic therapy could increase the particle tumor accumulation. Furthermore, Figure 5g, please provide statistical analysis for TUNEL and Ki67 staining (n=4) not just showing the representative images.

7. P21 “The gene expression profiles were tested by using an Affymetrix GeneChip PrimeView™ human gene expression array...” The samples are murine samples and cannot be used for human gene array. Please re-do the gene expression analysis using mouse genome arrays (e.g., Mouse Genome 430 2.0 Array or other similar murine genome arrays).

8. TOC just shows the chemical structure of M. The use of light and the incorporation of M into nanostructure particles are also important. Please redesign TOC to better illustrate the work.

Point-by-point responses to the review comments:

1. *Reply to the first comment made by Referee 1 “Please briefly comment on the advantages and novelty of this particular system over other phototheranostic nanoparticles.”*

This work exhibits unparalleled advantages and novelties listed as follows:

I). Most interestingly, chemotherapeutic drug (**cPt**) and photosensitizer (**TPP**) are used as building blocks to construct the metallacage, acting as the vertices and faces. Most photosensitizers are aggregated in aqueous solution due to their strong π - π stacking interactions, greatly limiting their applications in PDT. In this work, the π - π stacking of **TPP** was significantly inhibited by the formation of **M**, which is favorable to increase its fluorescence quantum yield and $^1\text{O}_2$ generation quantum yield. *In vitro* studies indicated that the metallacage-loaded nanoparticles (**MNPs**) were excellent phototherapeutic agents with a phototoxicity index as high as 246. Moreover, the anticancer efficacy of the platinum-based drug (**cPt**) was effectively maintained. Indeed, *in vitro* and *in vivo* studies confirm synergistic effect and superior anti-tumor performance of this smart supramolecular strategy.

II). The faces of the metallacage are suitable platforms to chelate Mn and ^{64}Cu ions, allowing the implementation of highly effective imaging techniques, such as magnetic resonance imaging (MRI) and positron emission tomography (PET) imaging. The simultaneous use of highly sensitive and high-resolution multimodal imaging methods helps to overcome the limitations of each modality independently and offers complementary and accurate insights into drug delivery, pharmacokinetics, accumulation, and excretion, which are highly desired in cancer theranostics.

III). Our design endows the resultant nanomedicine (**MNPs**) with targeting ability benefiting from EPR effect and active targeting capability, which are highly desirable to reduce side effects and improve therapeutic index. In this work, **mPEG-*b*-PEBP** and **RGD-PEG-*b*-PEBP** were chosen to PEGylate **M**, showing much higher loading content and stability than other amphiphilic diblock copolymers including PEG-*b*-PLA, PEG-*b*-PCL, PEG-*b*-PLGA, and PEG-*b*-PBLG. Different from other hydrophobic segments, the polyphosphoesters backbone can effectively avoid non-specific adsorption and uptake, which is favorable for extending the circulation time and tumor accumulation of **MNPs**. Indeed, PET imaging and bio-distribution studies demonstrated that high tumor accumulation of **MNPs** was achieved through our rational design. Furthermore, the high flexibility and hydration of the polyphosphoester backbones are favorable for water molecules to access the inner Mn ions thus maintaining the high r_1 value of the MRI agent by shortening their spin-lattice relaxation time. Compared with the clinically approved T_1 -contrast agent Magnevist, **MNPs** exhibit higher r_1 value and tumor accumulation.

IV). The combination of chemotherapy and PDT exhibits superior anticancer outcomes in combating against U87MG, drug-resistant A2780CIS, and orthotopic (breast and hepatoma) tumor models with negligible systemic toxicity, effectively preventing tumor recurrence and metastasis after a single treatment.

2. *Reply to the second comment made by Referee 1 “Please revise the schematic diagrams for improved simplicity and clarity for readers. For example, in Figure 1 (a), labels should be added for every structure. In Figure 1(b), the logical relationship amongst the three components could be made clearer.”*

The corresponding corrections have been done. We redrew the cartoon and added labels in the Figure.

3. *Reply to the third comment made by Referee 1 “More detailed analysis should be added from figures S4 to S9 to demonstrate that the structures were successfully synthesized. In the captions, full names for the abbreviations should be given. For example, what does TPPN stand for?”*

The corresponding corrections have been done. GPC studies were carried out to confirm the successful syntheses of these diblock copolymers. As shown in Supplementary Figure 6 and Supplementary Figure 9, the molecular weight of the diblock copolymer was much higher than its initiators, which provided direct evidence for the ring-open polymerization. The corresponding data have been added into the Supporting Information. Additionally, the full name of **TPPN** has been added.

4. *Reply to the fourth comment made by Referee 1 “Detailed information should be described for the loading content calculation by UV. The content percent of every element in EDS study should be added.”*

The corresponding corrections have been made. The description about the calculation of loading content has been described in the Methods. Additionally, the content percent of the elements in EDS study has been added in the Supporting Information.

5. *Reply to the fifth comment made by Referee 1 “Please provide further evidence to demonstrate the serum stability of MNPs. Suggest to increase the percent of serum from 10% to 50% FBS for particle stability assessment.”*

The corresponding correction has been made, and the corresponding discussion has been added into the Supporting Information. As shown in Supplementary Figure 16, negligible changes in average diameter of **MNPs** were detected in PBS containing 10% or 50% FBS after incubation for different periods of time, demonstrating that **MNPs** were stable under physiological environment.

6. *Reply to the sixth comment made by Referee 1 “Please add results of 4h post injection in the text corresponding to Fig. 4e, etc. The authors should check their description on Page 15, line 7 and line 12 and Page 34, line 18. Thorough proofread is needed.”*

The corresponding correction has been made.

7. *Reply to the seventh comment made by Referee 1 “Need strong references to support the claim*

that the half-life time of 1.87 ± 0.3 h is excellent. In Figure. 5(a), full pK analysis is needed (e.g., R2, t_1 , t_2 , etc.).

The corresponding corrections have been made. According to the comments from Reviewer 2, we conducted the pharmacokinetic studies using four mice in each group and calculated the circulation time accordingly. The distribution half-life (t_1) and elimination half-life (t_2) of **cPt** and **MNPs** were also obtained using a two-compartment fitting model (Supplementary Figure 27 and Supplementary Figure 28). The t_1 and t_2 values of **MNPs** were calculated to be 0.44 h and 4.19 h, which were much higher than that of **cPt** ($t_1 = 0.23$ h, $t_2 = 1.38$ h), confirming that the circulation time of **MNPs** was improved by taking advantage of EPR effect and active targeting capability. Compared with some reported nanomaterials, such as PEG modified UCNPs, iron oxide NPs and graphene, the half-life of **MNPs** was comparable or even better than them (*ACS Nano* 2015, 9, 6655–6674). The corresponding data and discussion have been added into the Supporting Information, and the reference was cited. Additionally, we deleted “excellent” in the main text to make the discussion more accurate and rigorous.

8. *Reply to the eighth comment made by Referee 1 “Please explain why “0.3 W cm⁻² power density of 671 nm laser with 10 min duration” were the optimal parameters? ”*

The corresponding correction has been made. In order to realize satisfactory photodynamic therapeutic outcomes, the laser density was kept at 0.3 W cm⁻² and the irradiation time was 10 min for *in vivo* studies. Photothermal effect would occur if we extend the laser irradiation time and enhance the laser density, which would disturb the therapeutic results. On the other hand, the photodynamic therapeutic performance would be inadequate if the laser irradiation time was shortened and the laser density was decreased.

9. *Reply to the ninth comment made by Referee 1 “It is important to investigate the drug release behavior from the nanoparticles in serum containing media or in different pH condition.”*

The corresponding studies have been done and added in the Supporting Information (Supplementary Figure 23).

10. *Reply to the tenth comment made by Referee 1 “Ref 14 does not provide the right context to introduce PDT nor it is the first self-assembled nanoporphyrin agents. Suggest to either replace it with Nature Material 2011, 10, 324 or with another classic PDT agent review (Chem Rev 2010, 110, 2839-2857).”*

The corresponding references recommended by the reviewer have been added.

11. *Reply to the first comment made by Referee 2 “P5 “M was quantitatively obtained by stirring the mixture in acetone/water (v/v = 8:2) at 70 °C for 5 h. The well-defined signals in 1H NMR spectra support the formation of a discrete and highly symmetric assembly as a sole thermodynamic product (Supplementary Fig. S10) 22. ” In SI Fig. S10 the NMR was measured in CD3CN, but in the synthesis procedure in SI PS3 “Then M was dissolved in CD2Cl2 for*

characterization.” Please verify the NMR solvent for Fig. S10. Also please make comparison of M with the mixture of TPP and DSTP in 1 H NMR spectra as the authors did in ref.22-23. Current Fig. S10 is less informative to show the distinctive formation of M.”

The corresponding corrections have been done. CD₃CN was used for ¹H NMR studies. The solubility of **TPP** and **DSTP** was poor in acetonitrile, especially for **TPP**. The resonance signals related to the protons of **TPP** could not be monitored in ¹H NMR spectrum (Supplementary Figure 12b–12d). However, their solubility in acetonitrile was significantly improved by the formation of metallacage (Supplementary Figure 12a). The well-defined signals in ¹H NMR spectra support the formation of a discrete and highly symmetric assembly as a sole thermodynamic product (Supplementary Figure 13). In ³¹P{¹H} spectrum, two different doublets at 6.24 and 0.92 ppm with concomitant ¹⁹⁵Pt satellites were observed in the spectrum of **M** (Fig. 2b). The observation of two doublets is consistent with the symmetry-breakage that occurs when one carboxylate moiety and one pyridyl coordinates to each Pt center. Electrospray ionization mass spectrometry further supported the formation of **M** owing to the observation of mass fragments corresponding to intact cores. Two peaks at $m/z = 1158$ and $m/z = 1486$ were monitored (Supplementary Fig. S14), ascribed to the intact tetragonal prism core with charge states, [**M** – 5OTf]⁵⁺ and [**M** – 4OTf]⁴⁺, respectively. The isotopic resolution of each peak was in agreement with the corresponding theoretical isotopic distribution, indicating that **M** possessed the expected 1:2:4 ratio of building blocks. These characterizations demonstrated the successful preparation of **M**. The corresponding data and discussion have been added into the Supporting Information.

12. *Reply to the second comment made by Referee 2 “The authors may provide the yield information about M, and whether or not a separation / purification step is needed or not. The authors may also consider to provide information about the scale –up capabilities of M and MNP, which could be advantageous for future translational studies.”*

The formed metallacage was precipitated using diethyl ether and the precipitation procedure was repeated four times to purify the metallacage with a yield of 91.6%. Notably, the synthesis of **M** could be easily scaled up, which was favorable for future translational studies. The corresponding discussion has been added into the Supporting Information.

13. *Reply to the third comment made by Referee 2 ““Metallocage” or “Metallacage” ? (both terms were seen in the literature).”*

The meaning of these two words is same. In our manuscript, we use “metallacge”.

14. *Reply to the fourth comment made by Referee 2 “P10 “A control experiment used TPPNPs fabricated from TPPN, mPEG-b-PEBP and RGD-PEG-b-PEBP with a loading content of 28.3%.” Please give the full name of TPPN and note it is different from TPP (the reader may think the TPPNP is a particle based on TPP instead of TPPN). It is also necessary to give the chemical design justification about TPPN (at least in SI). As the metallocage M is based on TPP and DSTP (benzene ring), why using naphthalene ring for TPPN synthesis instead of benzyl*

bromide? In addition, as TPPNPs were used for in vivo studies in the manuscript, it would be better to mention the use of TPPNP for photodynamic therapy as a control regimen at the first place.”

The corresponding description and discussion have been added into the main text and Supporting Information. To improve the solubility in organic solvent and increase the loading efficiency of the photosensitizer, **TPP** was modified using 2-(bromomethyl)naphthalene and the counterions were exchanged into hexafluorophosphate. Additionally, the full names of the diblock copolymers and **TPPN** have been provided.

15. *Reply to the fifth comment made by Referee 2 “P17 “...in order to extrapolate their pharmacokinetics (pKa).” It is unusual that “pharmacokinetics” is abbreviated to “pKa ” (often referred to acid dissociation constant). Please revise the abbreviation.”*

The corresponding corrections have been done. We changed “pK_a” into “pharmacokinetics” in the manuscript.

16. *Reply to the sixth comment made by Referee 2 “P20 Figure 5a “n=3.” At least n should be 4, and please compare the two groups statistically. Additionally, Figure 5b the group MNP is with light irradiation or not? Please compare intratumoral Pt accumulation for MNPs with and without light irradiation to see whether photodynamic therapy could increase the particle tumor accumulation. Furthermore, Figure 5g, please provide statistical analysis for TUNEL and Ki67 staining (n=4) not just showing the representative images.”*

The corresponding corrections have been done. We conducted the pharmacokinetic studies using four mice in each group and calculated the circulation time accordingly. Moreover, the distribution half-life (t_1) and elimination half-life (t_2) of **cPt** and **MNPs** were determined using a two-compartment fitting model (Supplementary Figure 27 and Supplementary Figure 28) according to the comment from Reviewer 1. The t_1 and t_2 values of **MNPs** were calculated to be 0.44 h and 4.19 h, which were much higher than that of **cPt** ($t_1 = 0.23$ h, $t_2 = 1.38$ h), confirming that the circulation time of **MNPs** was improved by taking advantage of EPR effect and active targeting capability. Compared with some reported nanomaterials, such as PEG modified UCNPs, iron oxide NPs and graphene, the half-life of **MNPs** was comparable or even better than them (*ACS Nano* 2015, 9, 6655–6674). We tested the intratumoral Pt amount with and without laser irradiation, but no significant difference was observed, indicating that laser irradiation could not affect the tumor accumulation of **MNPs**. Moreover, the apoptosis and proliferation percentage of the tumor cells for the mice after different treatments were calculated according to TUNEL and Ki67 staining (Supplementary Figure 32). The corresponding data and discussion have been added into the Supporting Information, and the reference was cited.

17. *Reply to the seventh comment made by Referee 2 “P21 “The gene expression profiles were tested by using an Affymetrix GeneChip PrimeView™ human gene expression array...” The samples are murine samples and cannot be used for human gene array. Please re-do the gene expression analysis using mouse genome arrays (e.g., Mouse Genome 430 2.0 Array or other similar murine*

genome arrays).”

In gene expression analysis, we chose human gene expression array to study the genetic changes after different treatments as the tumor cell lines are of human origin. Indeed, this method can provide valuable insight into the anticancer mechanism of each therapeutic modality (*Nat. Mater.* **2016**, *15*, 1128–1138; *Nat. Commun.* **2015**, *6*, 7939).

18. *Reply to the eighth comment made by Referee 2 “TOC just shows the chemical structure of M. The use of light and the incorporation of M into nanostructure particles are also important. Please redesign TOC to better illustrate the work. ”*

The corresponding correction has been done.